# Tribo-electrochemistry induced artificial solid electrolyte interface by self-catalysis

Chichu Qin[1,2], Dong Wang [1,2], Yumin Liu[1], Pengkun Yang[1], Tian Xie[1], Lu Huang [1✉], Haiyan Zou[1], Guanwu Li[1] & Yingpeng Wu [1✉]

Potassium (K) metal is a promising alkali metal anode for its high abundance. However, dendrite on K anode is a serious problem which is even worse than Li. Artificial SEI (ASEI) is one of effective routes for suppressing dendrite. However, there are still some issues of the ASEI made by the traditional methods, e.g. weak adhesion, insufficient/uneven reaction, which deeply affects the ionic diffusion kinetics and the effect of inhibiting dendrites. Herein, through a unique self-catalysis tribo-electrochemistry reaction, a continuous and compact protective layer is successfully constructed on K metal anode in seconds. Such a continuous and compact protective layer can not only improve the $K^+$ diffusion kinetics, but also strongly suppress K dendrite formation by its hard mechanical properties derived from rigid carbon system, as well as the improved $K^+$ conductivity and lowered electronic conductivity from the amorphous KF. As a result, the potassium symmetric cells exhibit stable cycles last more than 1000 h, which is almost 500 times that of pristine K.

---

[1] State Key Laboratory of Chem/Bio-Sensing and Chemometrics, Advanced Catalytic Engineering Research Center of the Ministry of Education, College of Chemistry and Chemical Engineering, Hunan University, 410082 Changsha, P. R. China. [2]These authors contributed equally: Chichu Qin, Dong Wang. ✉email: luhuang@hnu.edu.cn; wuyingpeng@hnu.edu.cn

The potassium (K) element owns abundant content in the earth's crust (~2.09 wt%, vs. ~0.0017 wt% of Li)[1–5] and low chemical potential (−2.93 V vs. the standard hydrogen electrode, SHE)[6–8]. However, the native SEI (solid electrolyte interface) of K is usually unstable for repeated cracks and reconstruction[9]. As a result, the K will stack as discrete nanoscale particles and grow along the pores of the separator, causing a rapid short circuit[9]. Thereof, SEI modification route by increasing Young modulus and ionic conductivity has been studied in recently[10,11]. Among those works, artificial SEI (ASEI) without electrolyte inducing has become a research hotspot[12], by which the functional elements in SEI can be precisely designed and controlled to effectively improve the ion transfer rates at interfaces. Meanwhile, ASEI can also protect highly reactive alkali metals and improve electrode stability. eg., commercial carbon paper[13], freestanding CNT film[14], and K–Hg alloy layer[15], which are usually directly covered on/react with the solid alkali metal. However, the interface issues between the electrode and ASEI are often been unsolved including weak adhesion or insufficient/uneven reaction, which significantly affects the ionic-diffusion kinetics and causes the localized growth of dendrites[16].

Recently, electrocatalysis is favored because of its ability to reduce the reaction activation energy significantly[17]. Commonly, contact electrification involves the transfer of charge between two surfaces, and it can give rise to surface charge densities[18] on the order of $10^{-5}$–$10^{-4}$ C m$^{-2}$. As reported, such charges can act as catalyst for the reaction on the surface[19]. On basis of contact electrification exhibiting the reaction between metals and triboelectric polymers[20,21], it is of great promise to get ASEI on the surface of the alkali metal.

In this study, we are first using a self-catalyzed tribo-electrochemistry reaction to construct a continuous and compact protective ASEI on potassium anode. During the reaction, the surface of K can transform to liquid to level the electrode and deepen the reaction. At the same time, the uniform triboelectric charge distribution can lead to a continuous and compact reaction layer, which is mainly composed of rigid carbon chain and amorphous KF. The hard mechanical properties derived from a rigid carbon structure can strongly suppress K dendrite formation. Amorphous KF can not only enhance the stability of electrode, but also improve the K$^+$ conductivity and lower the electronic conductivity of CCPP, which leads to larger and more uniform K deposited underneath the SEI. Therefore, the electrochemical performance of CCPP symmetric cell can be improved to more than 500 times that of pristine K. Meanwhile, the corresponding full cell (PTCDA as a cathode) also exhibits significantly long-term cycling stability and high rate capability.

## Results

Polytetrafluoroethylene (PTFE), as a strong electronegative material in the triboelectric series[22], was selected for the tribo-electrochemistry reaction with K metal. As we all know, PTFE has unusual chemical resistance[23] and only reacts with molten alkali metals and gaseous fluorine at high temperature and high pressure. However, as shown in Supplementary Movie 1 and Fig. 1a, an explosive reaction happens accompanied by the liquefaction of K and splashing of products when only pressure is applied. Based on this phenomenon, continuous and compact protected potassium (CCPP) anodes were prepared.

**Propose to the "Self-catalysis tribo-electrochemistry" mechanism**. It has been reported that LiF/defluorinated polymers SEI can be obtained by continuous roll-press[24], but the reaction mechanism is still not clear. Differently, in our case, only an

inducing force is needed to start the reaction without any other physical and chemical processes.

As shown in Fig. 1b, based on the triboelectric series of different capabilities to obtain electrons[25], a mechanism of the ASEI formation is proposed, namely a self-catalysis tribo-electrochemistry reaction: (1) a force working as the initiator to charge accumulation; (2) charge (electric field) working as the trigger (catalyzer) to induce the initial defluorination accompanied by release heat; (3) the accumulation of the reaction heat induced by the ongoing reaction; (4) liquefaction of K metal by heat for deeper reaction; (5) propagation reaction induced by the increased heat for the ASEI. Indeed, if the accumulation of charge exceeds a certain critical point, an "explosive" reaction (Fig. 1a and Supplementary Movie 1) will occur.

**Four progressive parts to clarify the reaction mechanism**. First, a trigger (catalyzer) is needed.

Typically, solid K metal does not react directly with PTFE. There are no significant changes within a brief period, either in K metal or PTFE film, when the two just contact with each other without other external forces (Supplementary Fig. S1). However, when a force (including press, friction, etc.) is applied, with the increasing of the force, a critical point can be reached where an instantaneous and powerful reaction happens, with sparks, smoke, and pulverization of alkali metals and PTFE (Supplementary Movie 1, Fig. 1a, and Supplementary Fig. S2a, b). On the contrary, a very slow reaction happens between PTFE and liquid Na–K alloy without force (Supplementary Fig. S3b), while quick and violent reaction occurs between PTFE and liquid Na–K alloy with a certain friction or pressure (Supplementary Movie 2 and Supplementary Fig. S3a). These phenomena indicate that something critical triggers and accelerates the reaction process, we call it the trigger (catalyzer).

Second, what's the trigger (catalyzer)?

No violent reactions occur when the liquid Na–K alloy droplet simply touches the PTFE plate. However, by repeatedly lifting and dropping the liquid Na–K droplet onto the PTFE plate, an explosive reaction happens after several cycles (Supplementary Movie 3 and Supplementary Fig. S4). This phenomenon means something accumulates during the lifting and dropping cycles and reaches a critical point when the reaction happens.

Recently, Wang and co-authors reported a high-density charge induced on the surface of PTFE film under the impinging water droplets[26]. Based on it, it can be speculated that charges accumulate on PTFE during the lifting/dropping process and trigger/catalyze the reaction. To further confirm this, grounding experiments[27] were performed (Supplementary Fig. S5a). When grounded, charges cannot accumulate on the interface between PTFE and metal, and no explosive reaction happens during the repeated tapping or rubbing (Supplementary Fig. S5b). Whereas, once without grounding, there will be an immediate spark even only a little friction is applied (Supplementary Fig. S5c).

For direct proof, charge measurement is made with the grounding device (Supplementary Fig. S5d). As shown in Fig. 1c, no current can be detected when no force is applied to the joint surface between PTFE and K metal (blue line, baseline). However, as the regular pressure is applied, the current output signal will appear correspondingly (red line). Supplementary Fig. S5e excludes the interference of environmental forces on the current output signals.

Taking these into consideration, we can conclude that the powerful reaction between solid K and PTFE can be triggered at room temperature when the negative charges accumulate on PTFE. Once pressure is applied, a lateral shear[28] occurs between K and PTFE. That is, there is a friction interaction between the

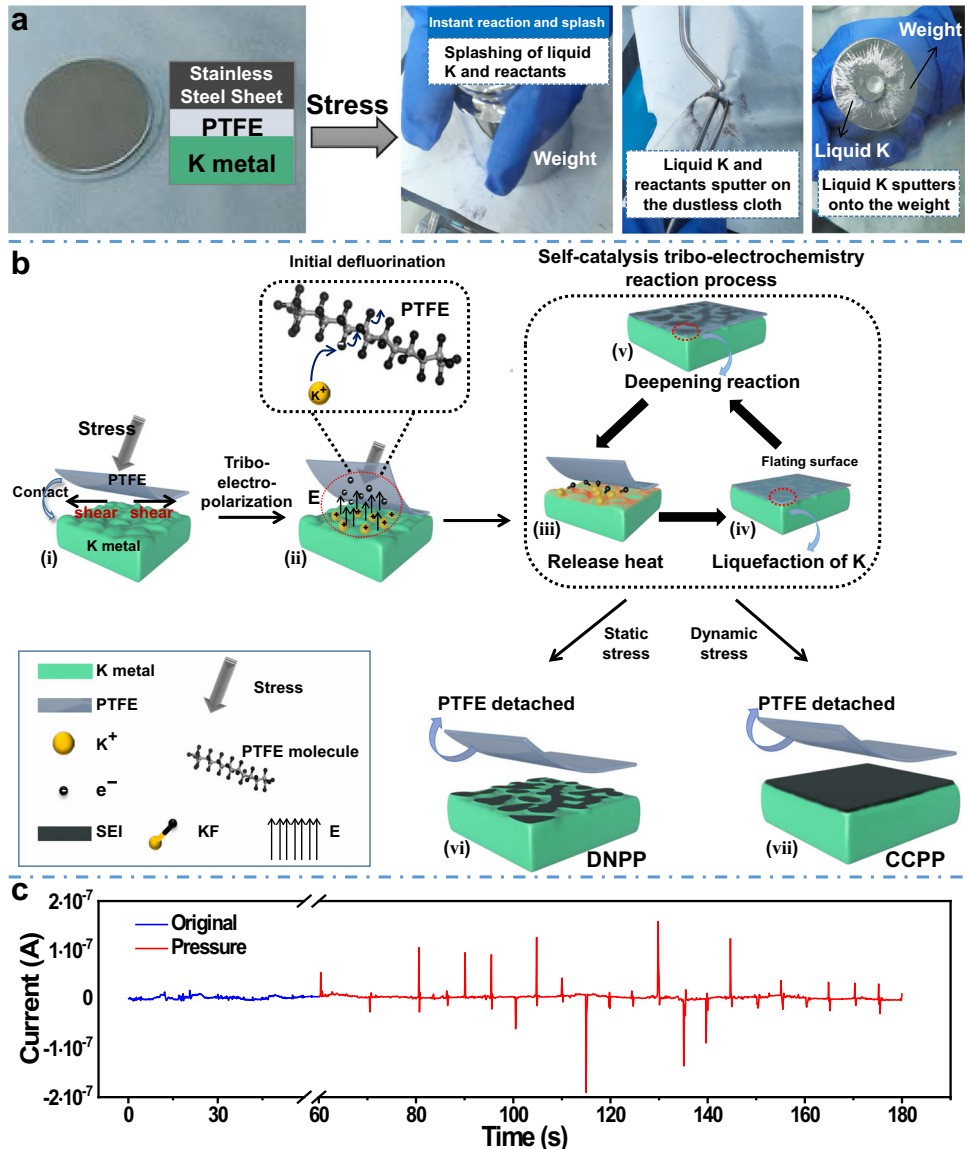

**Fig. 1 Self-catalysis tribo-electrochemistry mechanism and related experiments. a** Schematic diagram of explosive reaction and K liquefaction when strong pressure is applied to the joint surface between PTFE and K metal. **b** Schematic diagram of self-catalysis tribo-electrochemistry reaction process. **c** The output current with regular force applied to the grounded joint surface between PTFE and K metal. Source data are provided as a Source Data file.

two, causing the transfer of charge between each other, as shown in Fig. 1b(i). The extra electrons lead to a high triboelectric field (Fig. 1b(ii)), acting as the catalyzer and triggering the reaction. And external force is only the initiator to the charge.

Third, the initial defluorination is triggered/catalyzed by the accumulated charges.

With the accumulated charges, an electric field is built. Promoted by the electro-inductive effect[29] from the electric field, the F atoms begin to react with K[23] at room temperature, and the C–F bond will be broken down as well as the KF will be generated[24], as shown in Fig. 1b(ii) and Eq. (1).

$$-(CF_2 - CF_2)_n + K \rightarrow -(CF_2 - CF_2)_x - CF = CF - (CF_2 - CF_2)_y + KF$$

$$\Delta H_F \approx -1421\,KJ \cdot mol^{-1}$$

$$(1)$$

The initial defluorination of PTFE by reacting with 1 mol K will release 1421 KJ mol$^{-1}$ heat (Fig. 1b(iii)), which is theoretically enough to melt ~1308 mol K[30].

Fourth, the liquefaction of K is the key factor to flat the K surface and get the uniform and dense ASEI.

The surface of solid K metal is uneven and has many bumps (Supplementary Fig. S6). Liquid K has better fluidity, wettability, and permeability. Once the reactions between PTFE and K begin, K protrusions will be melted by reaction heat and liquid K will spread out quickly to level the K metal surface, as shown in Fig. 1b(iv).

Solid K will hardly react with PTFE without the catalysis of charge, which is evidenced by our experiments (Supplementary Fig. S1). At the same time, it has been reported that liquid K (including K amalgam[31] and hot melt K[32]) can react with PTFE. We also found that the liquid Na–K alloy can react spontaneously with PTFE slowly (Supplementary Fig. S3b). Thereof, after the initial defluorination, the released heat and liquefied K will make the reaction extremely easier, wider, and deeper, as is presented in Fig. 1b(v). Then more heat released and more K liquefied, the reaction is further deepened. That is, a chain reaction begins (Fig. 1b(iii, iv, v)), leading to a continuous and uniform reaction and a compact reaction layer (Fig. 1b(vii)). If the charge cannot be

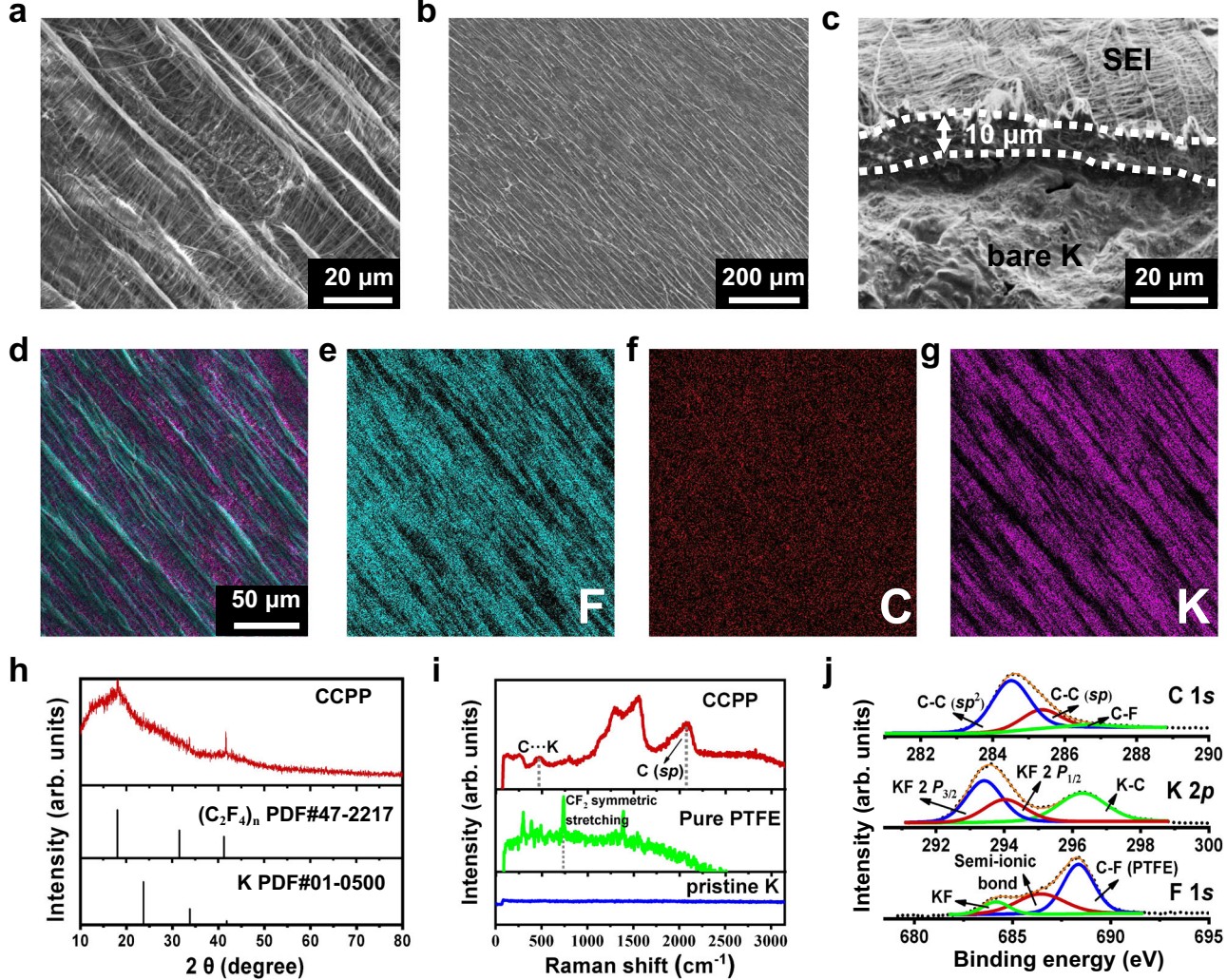

**Fig. 2 Morphologies and composition characterizations of CCPP. a, b** Top views of CCPP by SEM. **c** Cross-sectional view of CCPP, the thickness of the SEI is ~10 μm. **d–g** The corresponding elemental mappings of CCPP. **h** XRD pattern of CCPP. **i** Raman spectrums of CCPP, pristine K, and Pure PTFE. **j** C 1s, K 2p, and F 1s XPS spectra of CCPP. Source data are provided as a Source Data file.

generated continuously, a discontinuous and non-compact reaction layer will be covered (Fig. 1b(vi)).

**Self-catalysis tribo-electrochemistry for artificial SEI.** In order to get the continuous and compact protected potassium (CCPP) anodes, a piece of solid-state PTFE film is covered on the K metal. Repeated friction or pressure is then applied to the plane of PTFE to induce an ongoing self-catalytic reaction between K metal and PTFE. After the full reaction, the residual PTFE precursor is detached, and then a continuous and compact black protective layer is formed (Supplementary Fig. S7b). In contrast, if static stress is applied, the charge cannot be continuously generated, and the discontinuous and non-compact protected potassium (DNPP) is obtained (Supplementary Fig. S7a).

**Characterizations of ASEI.** Different from the rough surface of pristine K (Supplementary Fig. S6), Scanning electron microscope (SEM) images show flat surfaces of CCPP (Fig. 2a, b), the cross-sectional view further reveals that a continuous and compact SEI tightly adheres to the surface of the K metal (Fig. 2c). In addition, corresponding elemental mappings show the homogeneous F, C, and K distribution (Fig. 2d–g and Supplementary Figs. S8 and S10a–d). All of these can be attributed to the leveling effect and deepening reaction in the self-catalysis tribo-electrochemistry

cycles. The SEI of CCPP with dense character and reasonable thickness is essential to enhance the ion-diffusion kinetics and suppress the growth of K dendrite[16]. In contrast, the surfaces of DNPP anode show discontinuous (Supplementary Fig. S9a, b) and non-compact (Supplementary Fig. S9c) ASEI. Energy dispersive X-ray spectroscopy (EDS) analysis (especially F element) further confirms the non-uniform distribution of the SEI of DNPP, as shown in Supplementary Fig. S10e–h.

The X-ray diffraction (XRD) spectra reveal that the components of CCPP' SEI are approaching an amorphous state (Fig. 2h and Supplementary Fig. S11), which benefits for larger and more uniform K plating[33]. The results of the Raman characterization (Fig. 2i and Supplementary Fig. S12) confirm the existence of graphite carbon[34] (D peak at 1337 cm$^{-1}$, G peak at 1542 cm$^{-1}$) and sp C (2100 cm$^{-1}$)[35] in the SEI of CCPP anodes. Meanwhile, a strongest Raman peak at 734 cm$^{-1}$ (represent CF$_2$ symmetric stretching[21]) is easily visible in PTFE, but absent in the CCPP, which further confirming that an adequate defluorination reaction occurs between K and PTFE. The Raman results indicate that the ASEI contains carbon with rigid structure[36,37] (including graphite and possibly graphdiyne[38] or carbine[39], etc.), as well as a small amount of K–C bond. It is worth noting that such a rigid carbon structure can ensure the CCPP anodes high mechanical properties and conductivity[40]. Stress–strain curves confirm that

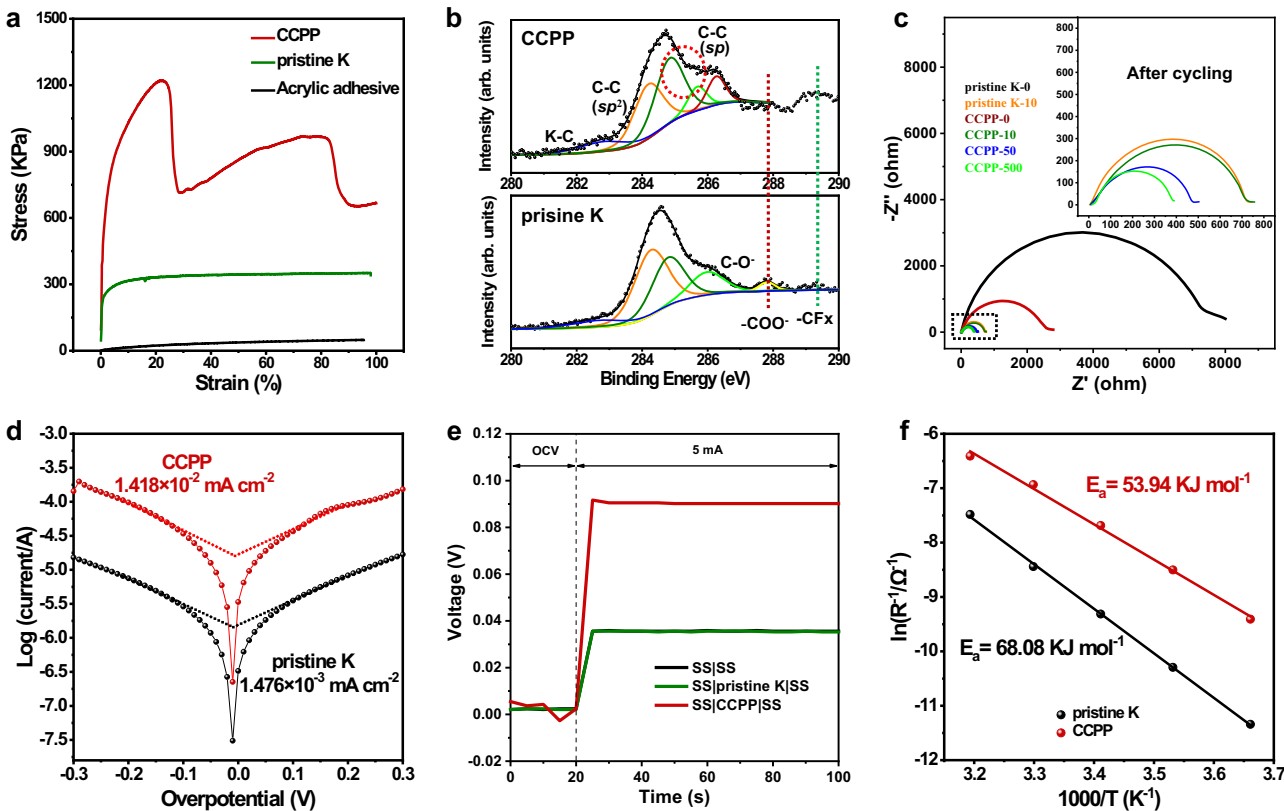

**Fig. 3 Mechanical, interfacial and ion-transport properties of CCPP. a** Stress–strain curves of CCPP, pristine K, and acrylic adhesive. **b** C 1s XPS spectra of CCPP and pristine K after 20-h cycling. **c** Nyquist plots taken at various cycle numbers. **d** Tafel plots and derived exchange current density obtained from linear sweep voltammetry tests in the range of −0.3 V to 0.3 V at 2 mV s$^{-1}$. **e** D.c. conductivity Measurements of pristine K and CCPP using blocking electrodes. **f** Corresponding Arrhenius curves and comparison of activation energies of CCPP and pristine K. Source data are provided as a Source Data file.

the mechanical strength of CCPP is about four times that of pristine K (Fig. 3a). The native SEIs of K formed in the electrolyte are fragile and prone to crack during cycling[9]. Therefore, the greatly improved mechanical strength and hardness of CCPP can strongly suppress K dendrite formation[12]. In order to probe more detailed chemical information about the ASEI composition, XPS tests were performed without air contact (Fig. 2j). In detail, the C 1s spectrum can be divided into three peaks, including the C–C bond ($sp^2$) at 284.5 eV, C≡C bond ($sp$) at 285.3 eV and C–F bond at 286.7 eV[41]. In F 1s spectrum, the signals of KF (684.2 eV), nearby semi-ionic fluorinated carbon chain (686.5 eV) and covalently bonded fluorine atoms (PTFE, 688.4 eV) can be characterized[41]. Combined with the K 2p spectrum, the main components of the CCPP also have KF $2p_{3/2}$ (293.6 eV), KF $2p_{1/2}$ (294.08 eV), and K–C bond (296.36 eV)[9]. Based on these results, the SEI of CCPP is mainly composed of amorphous KF, and rigid carbon chain (C–C bond mainly in the graphite carbon, C≡C as well as a small amount of C–F bond and K–C bond). Thus, the reaction equation of K metal and PTFE can be summarized as shown in Eq. (2):

$$-(CF_2 - CF_2)_n - + 4n\,K \rightarrow 2n\,C + 4n\,KF \qquad (2)$$

In order to explore the chemical stability of the SEI on the surface of CCPP electrode, Raman and XPS spectra after cycling were further characterized. The results show that the CCPP maintains the original composition after cycling (Fig. 3b and Supplementary Fig. S14a), which indicates that it has good chemical stability and can effectively prevent the side reactions between electrolyte and the K metal, leading to a high coulombic efficiency (C.E.)[42,43]. On the contrary, on the surface of pristine K

without modification, electrolyte decomposition components can be detected after cycling, including K–C, C–O$^-$, –COO$^-$, –CF$_x$, etc. (Fig. 3b and Supplementary Fig. S14b). This result indicates that the SEI of pristine K is unstable and prone to rupture during repeated cycling, then, the electrolyte will decompose by side reactions with the freshly exposed K metal.

To further investigate the role of CCPP, electrochemical impedance spectroscopy (EIS) is conducted. Figure 3c shows two EIS plots taken at various cycle numbers, and the corresponding comparison of $R_{in}$ is shown in Supplementary Fig. S15b, c. The impedances of the SEI are calculated through the equivalent circuit fitting (Supplementary Fig. S15a and Supplementary Table S1). Obviously, the initial interface impedance ($R_{in}$) of pristine K (8337 Ω) is much higher than CCPP (2789 Ω), which reflects better K$^+$ migration kinetics[44] of the SEI of CCPP benefit from the continuous and compact feature[16]. In addition, both the $R_{in}$ of CCPP and pristine K show a significant decrease after ten cycles (Fig. 3c). However, the $R_{in}$ of pristine K drops quickly after ten cycles (Supplementary Fig. S15b and Supplementary Table S1), because of the formation of unstable and fragile SEI on the surface of bare K electrode, leading to rapidly and violently dendrites generation during cycling. The freshly exposed K metal dendrites immediately contact the electrolyte to form a thin SEI with larger conductivity (reasons for the rapid decline of $R_{in}$), which further intensifies the growth of dendrites, and the battery quickly failed. Unlike pristine K, the $R_{in}$ of CCPP shows a slower decreasing trend (Fig. 3c, Supplementary Fig. S15c, and Supplementary Table S1), because the SEI of CCPP possesses extremely strong stability[45] derived from the amorphous KF[43].

Tafel plots (Fig. 3d) confirm that the exchange current density of CCPP (0.01418 mA cm$^{-2}$) is one order higher than pristine K

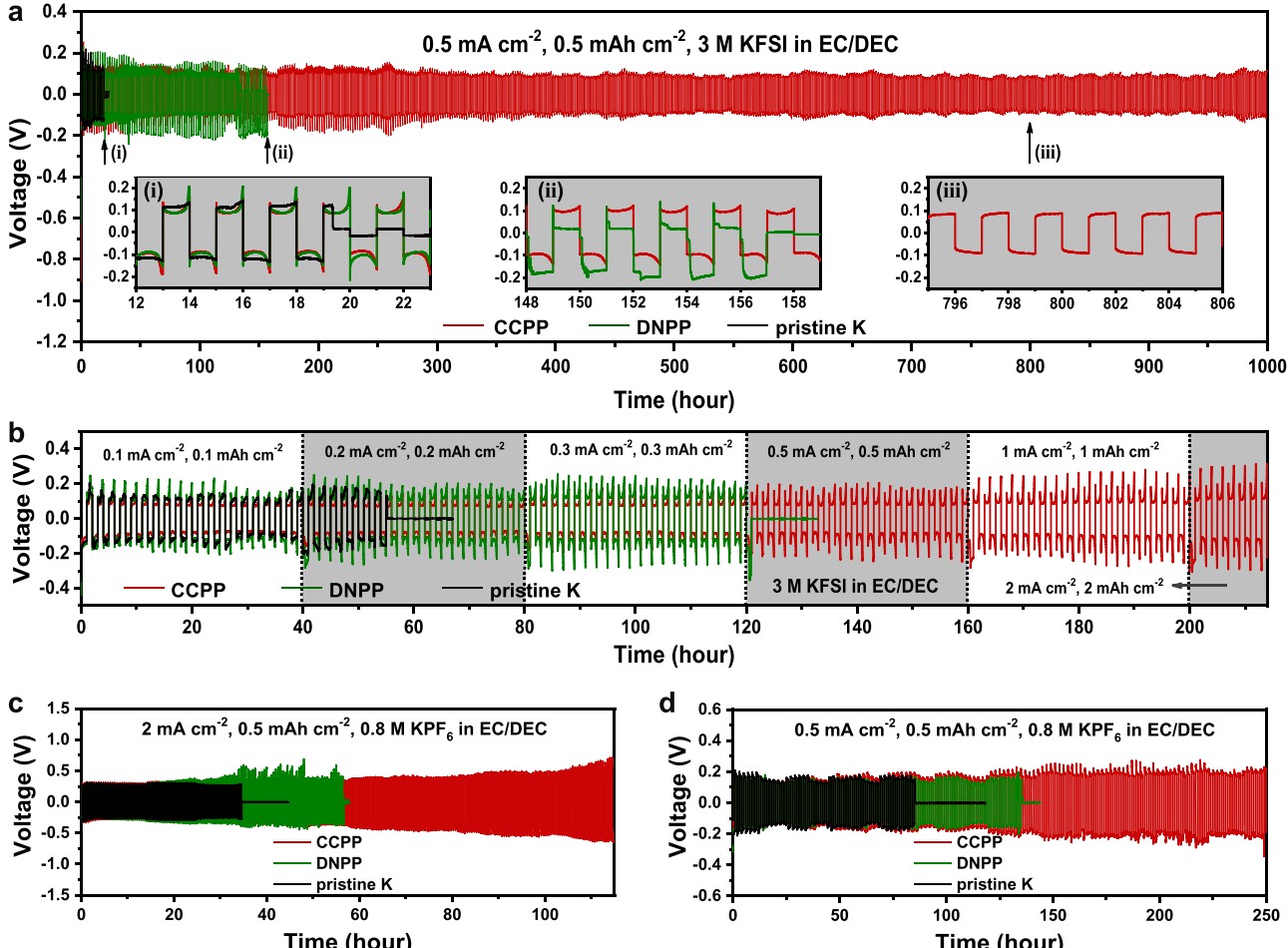

**Fig. 4 Electrochemical properties of CCPP. a** Voltage–time profiles of CCPP, DNPP and pristine K at the current density of 0.5 mA cm$^{-2}$ and capacity of 0.5 mAh cm$^{-2}$ in the highly concentrated KFSI/EC:DEC electrolyte, the illustrations (i), (ii), and (iii) are the detailed comparison of these three at 12–23 h, 148–159 h, and 795–806 h, respectively. **b** Rate performances of the three symmetric cells at different current and capacity conditions in the highly concentrated KFSI/EC:DEC electrolyte. **c** Polarizations of CCPP, DNPP and pristine K at 2 mA cm$^{-2}$ and 0.5 mAh cm$^{-2}$ by using KPF$_6$/EC:DEC electrolyte. **d** Polarizations of CCPP, DNPP and pristine K at 0.5 mA cm$^{-2}$ and 0.5 mAh cm$^{-2}$ by using KPF$_6$/EC:DEC electrolyte. Source data are provided as a Source Data file.

(0.001476 mA cm$^{-2}$). Combined with the impedance results, the sluggish reduction of interfacial resistance, as well as the high exchange current density, means higher potassium ion conductivity[46] and charge-transfer kinetics[16,47] in CCPP, which is mainly due to the amorphous KF and carbon[41,48]. The electronic resistivity of the SEI of CCPP is about $1.23 \times 10^4\ \Omega$ cm (Fig. 3e), which can facilitate the K plating underneath the SEI preferentially[16,49]. Meanwhile, the activation energy of CCPP was calculated as only 53.94 KJ mol$^{-1}$ (Fig. 3f, the temperature-dependent EIS and the fitting results of R$_{in}$ are shown in Supplementary Fig. S16 and Supplementary Table S2, respectively.), whereas pristine K showed much higher activation energy (68.08 KJ mol$^{-1}$). Such results reveal a lower reaction barrier and faster charge-transfer kinetics[50,51] of CCPP than pristine K resulting from rigid carbon structure[52] and amorphous KF[41,53].

**Electrochemical performances of the CCPP electrode.** In the highly concentrated KFSI/EC:DEC electrolyte, plating/stripping tests are carried out in K symmetric cells with the CCPP, DNPP, and pristine K anodes, respectively. Figure 4a shows the long-term galvanostatic cycling stability of the three anodes. The pristine K symmetric cell suddenly short circuits after 19 h, attributed to the deterioration of dendrites on the surface of the

unprotected K metal anode, which quickly penetrated through the separator. Compared with pristine K, the DNPP can run stably over 100 h due to a protective film, then the voltage polarization curve becomes unstable after 130 h and eventually short circuits at 157 h. This result may be due to the fact that the SEI of DNPP is not thick and continuous enough (Supplementary Fig. S9 and Supplementary Fig. S10e–h) to suppress the dendrite at a later stage. In contrast, the CCPP anodes can keep stable polarization for more than 1000 h (Fig. 4a and Supplementary Fig. S18a, b). Despite the large overpotential due to the thick SEI in the early stages (illustration (i) in Fig. 4a and Supplementary Fig. S17b), the voltage plateau gradually becomes flat and stabilizes at about 92 mV (illustration (i)-(iii) in Fig. 4a and Supplementary Fig. S17). Surely, residual unreacted PTFE on the SEI has been proven to be no contribution to the excellent performance for direct PTFE coverage can make the battery short circuit quickly (Supplementary Fig. S19). In addition, the symmetric cell with CCPP anode also has superior long-term cycling performance both at higher current density or higher capacity. Under the conditions of 1 mA cm$^{-2}$ and 1 mAh cm$^{-2}$, the CCPP electrodes are stable for more than 270 h with a voltage polarization of ~110 mV. By contrast, the pristine K electrode short circuits after 13 h (Supplementary Fig. S20a). Even under the high-capacity cycling conditions of 2 mAh cm$^{-2}$, the CCPP anodes is

stable for more than 140 h, while a pristine K metal anode can even not persist for one cycle (Supplementary Fig. S20b). CCPP anodes also offer significant advantages in terms of rate performance (Fig. 4b and Supplementary Fig. S22). As clearly shown in Fig. 4b, as the current density and cyclic capacity increases from 0.1 to 2 mA/mAh cm$^{-2}$, the CCPP cell delivered voltage hysteresis of 83, 81, 82, 88, 103, and 139 mV (Supplementary Fig. S21). In contrast, the cell with DNPP electrodes and pristine K metal anodes demonstrate an inferior tolerance: the DNPP electrodes cannot complete a stable leap from 0.3 mA cm$^{-2}$ to 0.5 mA cm$^{-2}$ and the pristine K anodes were even worse, shorting out in less than nine cycles at a current density of 0.2 mA cm$^{-2}$.

In addition, the CCPP anodes also exhibit remarkable superiority in KPF$_6$/EC:DEC electrolyte. Under a large current density of 2 mA cm$^{-2}$, maintaining a stable polarization for more than 110 h (Fig. 4c). In contrast, DNPP and pristine K become short circuited after 56 h and 35 h, respectively, due to the lack of stable and robust SEI protection. Figure 4d shows the voltage profiles at 0.5 mA cm$^{-2}$ and 0.5 mAh cm$^{-2}$, pristine K and DNPP short circuit ascribe to the dendrites within 90 h and 140 h, respectively. CCPP, by contrast, can run stably >250 h, which is almost three times that of pristine K. For the rate performance, CCPP can withstand circulation at higher current densities (0.1 mA cm$^{-2}$ to, 0.5, 1, 1.5, 2, and 3 mA cm$^{-2}$). Whereas, pristine K cannot last more than four cycles at 1.5 mA cm$^{-2}$ and 1 mAh cm$^{-2}$ (Supplementary Fig. S23). In a word, CCPP possesses better performances not only in the highly concentrated KFSI/EC:DEC electrolyte but also in KPF$_6$/EC:DEC electrolyte, which indicates that the importance of a continuous and compact robust SEI to inhibit K metal dendrites.

The morphological characterizations of the electrodes after cycling further confirmed the strong superiority of CCPP anodes over pristine K and DNPP electrodes. Figure 5c, d shows the cross-sections of CCPP anodes after 10 and 50 cycles, respectively. The CCPP anode maintains good compactness from surface to bulk phase after 10 cycles. Even when the cycle number is increased to 50, only a small thickness of cyclic K is uniformly deposited beneath the SEI (insert in Fig. 5d). Correspondingly, from the top view, the CCPP maintains complete and smooth surface states (Fig. 5g, h), even after 1000-h of cycling (Supplementary Fig. S24). These results can mainly attribute to the significantly enhanced mechanical properties of CCPP derived from the rigid carbon chain structure, which can strongly suppress the growth of dendrites. At the same time, the amorphous KF effectively improves the K$^+$ conductivity and lowers the electronic conductivity of CCPP, which guides K prior to plating underneath the SEI. In magnified SEM (Supplementary Fig. S25), the K metal uniformly distributes within the interior of the fibrous structure in a granular form. This behavior can be attributed to the evenly distributed glassy KF, leading to larger and more uniform granular K deposition[33]. In contrast, due to the incomplete SEI coverage on the surface of the DNPP electrode, dendrites grow from the cracks after cycling (Supplementary Fig. S26a, b, d). Due to the lack of stable SEI protection, pristine K is the worst and exhibits increased volume expansion after 10 and 50 cycles, as well as a noticeable roughness and unevenness of the surface (Fig. 5a, b). Particularly in the top view, untreated K anode appears to have heavy dendrite growth accompanied by a high level of powdered K (Fig. 5e, f and Supplementary Fig. S26c). Noticeably, after cycling under the same conditions, the separator in the CCPP battery still maintains good integrity (Supplementary Fig. S27c).

Different from our ASEI construction strategy, traditional ASEI often has some interface issues. Figure 5i illustrates three common types of ASEI: direct coating, "solid" to "solid" reaction, or "liquid" to "solid" reaction, which is prone to loose contact,

inadequate and uneven reaction. In contrast, the excellent performance of CCPP anodes can mainly be attributed to the unique self-catalysis tribo-electrochemistry SEI formation strategy. Due to the catalysis of the electric field, the surface K is transformed from solid to liquid to level the electrode and deepen the reaction, as well as the uniform triboelectric charge distribution brought by repeated friction, the CCPP is continuous and compact, which feature can significantly improve the K$^+$ migration kinetics (Fig. 5j).

Tests of full cells matched with PTCDA confirmed the feasibility of CCPP anodes in real applications. Figure 6a shows the long-term stability of CCPP||PTCDA and pristine K||PTCDA at the current density of 500 mA g$^{-1}$, the CCPP||PTCDA cell shows a discharge capacity of 133.56 mAh g$^{-1}$ (5th cycle) when stabilized for a number of cycles and keep a capacity of 117.5 mAh g$^{-1}$ after 200 cycles with a capacity retention of 88% (Fig. 6b). However, pristine K||PTCDA is broken soon after 31 cycles because of the intensified dendrites growth and electrolyte decomposition for the lack of stable SEI protection. Furthermore, the CCPP||PTCDA cell delivers reversible capacities of 138.8, 136.8, 134.6, 132.6, 130, 120 mAh g$^{-1}$ at current density of 30, 50, 100, 200, 500, 1000 mA g$^{-1}$, respectively (Fig. 6c and Supplementary Fig. S28). By contrast, pristine K||PTCDA cell delivers 138.4, 135, 131.5, 127.2, 115, 103.6 mAh g$^{-1}$, respectively. These results direct to superb rate capability of CCPP||PTCDA cell. In addition, CCPP||PB and pristine K||PB cells were also assembled and tested to verify the effects of the continuous and compact SEI. As shown in Supplementary Fig. S29, although both cells deliver a comparable initial capacity of about 77 mAh g$^{-1}$, the capacity of pristine K||PB drops to 67.7 mAh g$^{-1}$ after 150 cycles (Supplementary Fig. S29c). The CCPP||PB cell, in contrast, could delivers a higher capacity of 74.9 mAh g$^{-1}$ (Supplementary Fig. S29b), indicating reduced loss of electrolyte and capacity of CCPP||PB cell.

## Discussion

We have developed a self-catalyzed tribo-electrochemistry strategy to construct a continuous and compact protective layer on the K electrode surface. The unique SEI construction strategy can be finished in seconds and greatly solve the interface issues between "traditional" ASEI and electrode surfaces with no complicated physical and chemical methods. In addition, the obtained CCPP anode exhibited prominent electrochemical performance enhancement both in symmetrical battery and full battery. That is, the potassium symmetric cells exhibit stable cycles last more than 1000 h at long-term symmetric cycling, which is over 500 times that of pristine K. Meanwhile, the corresponding K||PTCDA full cell also exhibits significantly stable longer duration (88% capacity retention for 200 laps at 500 mA g$^{-1}$ in the KPF$_6$/EC:DEC Electrolyte) and high rate capability (still has a capacity of 119.7 mAh g$^{-1}$ at 1000 mA g$^{-1}$). We anticipate that the self-catalyzed tribo-electrochemistry strategy will open a new thoroughfare for the protection of high-energy-density alkali metal anodes.

## Methods

**Materials.** Perylene-3, 4, 9, 10-tetracarboxylic dianhydride (PTCDA, C$_{24}$H$_8$O$_6$, 98%, the commercial PTCDA are consistent with the crystal structure of PTCDA[54] as Supplementary Fig. S30), and Potassium (K, 99%) are provided by Aladdin Reagent and used as received without further purification. FeCl$_3$ (CP, ≥ 97.0%) and K$_4$Fe(CN)$_6$·3H$_2$O (AR, ≥ 99.5%) are provided by China National Medicines Co., Ltd. and used as received without further purification. Super P, acetylene black, carbon-coated aluminum foil, and PTFE condensed liquid binder are purchased from Shenzhen Kejing Star Technology Co., Ltd. The PTFE film (20 μm) is obtained from Shenzhen Gorest Technology, China. All the electrolytes are purchased from Suzhou Dodochem Ltd., China.

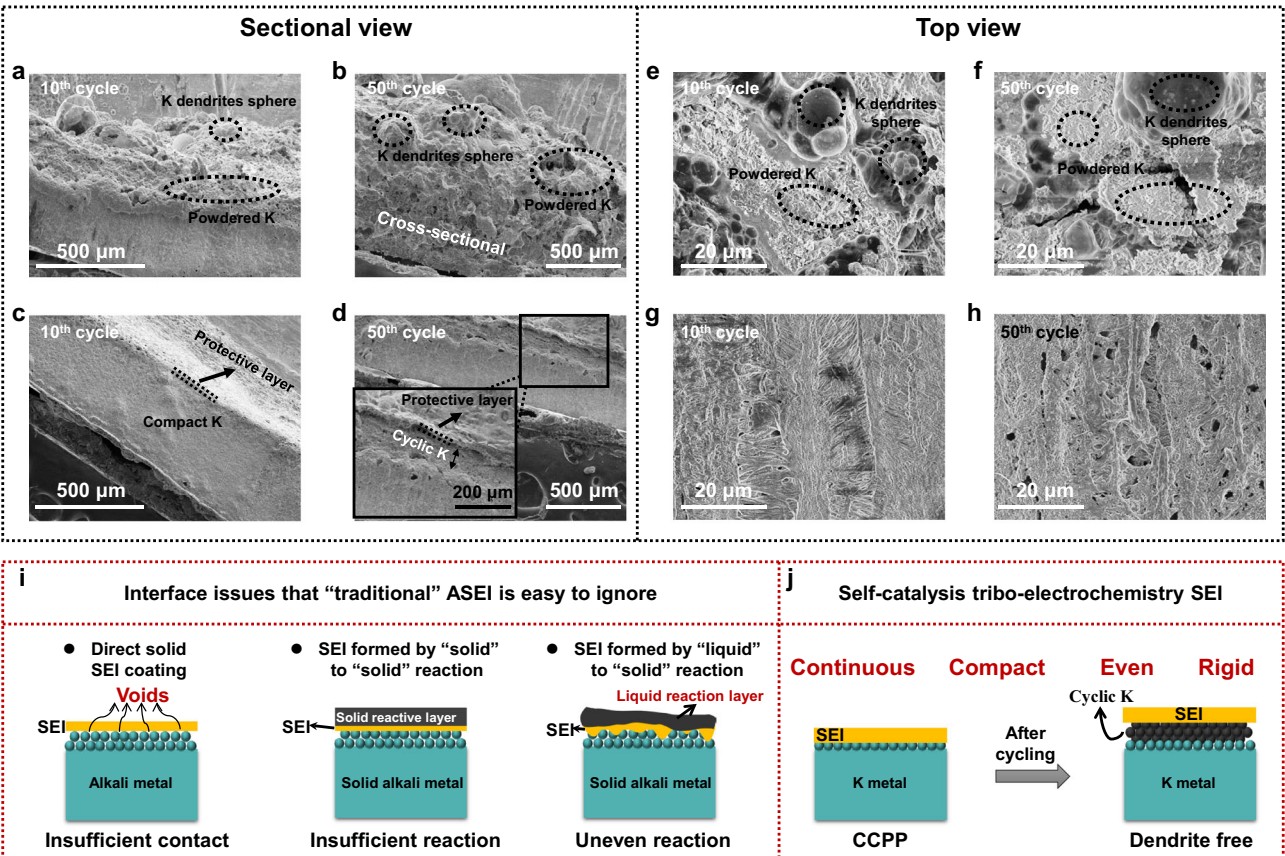

**Fig. 5 SEM images of CCPP and pristine K after cycling and the advantages of SEI constructed by self-catalysis tribo-electrochemistry reaction. a**, **b** Cross-sections of pristine K after (**a**) 10 and (**b**) 50 loops. **c**, **d** Cross-sections of CCPP anode after (**c**) 10 and (**d**) 50 loops. **e**, **f** Top views of pristine K after (**e**) 10 and (**f**) 50 loops. **g**, **h** Top views of CCPP anode after (**g**) 10 and (**h**) 50 loops. **i** Interface issues often ignored by three "traditional" ASEI construction strategies. **j** Inhibition of dendrites by CCPP electrode, which SEI constructed by self-catalysis tribo-electrochemistry reaction.

**Pristine K anode preparation**. All the K foils are pressed on stainless steel sheets (for electrochemical measurements) or Cu foil (for SEM, Raman or XPS characterization) with a diameter of 12 mm in a glove box filled by Ar gas (>99.99%) within the $O_2$ and $H_2O$ less than 0.1 ppm. The surface of K foil is cleaned artificially through scraping until a shiny metallic surface appeared, as shown in the illustration of Supplementary Fig. S6a. Commonly, the mass of the potassium foil is about 0.055 g.

**CCPP anode preparation**. As shown in Supplementary Fig. S7b, the continuous and compact potassium (CCPP) anode is prepared by a self-catalyzed tribo-electrochemistry strategy. That is, a piece of PTFE film is firstly stacked on top of the K foil so that their surfaces are in good contact. Friction or pressure is then repeatedly applied (dynamic stress) to the joint surface until an intact black film appeared on the K foil surface. By contrast, under static stress, a discontinuous and non-compact protective potassium (DNPP) anode is prepared (Supplementary Fig. S7a).

**Preparation of PTCDA electrode**. The freestanding PTCDA electrode is comprised of 60 wt% PTCDA powder, 30 wt% Super P, and 10 wt% PTFE binder. First, PTFE condensed liquid binder and deionized water are mixed with a mass ratio of 1:99 and then ultrasound 30 min to obtain PTFE aqueous solution with a mass fraction of 0.6%. Second, 0.1 g PTCDA and 0.05 g Super P are weighed, subsequently, the absolute ethyl alcohol is added to 1/3 of the reagent bottle (20 mL), and then sonicated for 30 min to obtain the precursor solution. Third, 2.778 g of PTFE aqueous solution with a mass fraction of 0.6% are added to the precursor solution with ultrasonic power at about 1 h. Finally, the mixed solution is dried in the oven at 60 °C for 8 h. After the sample is completely dry, wet it with absolute ethyl alcohol and press repeatedly to make it fully bonded. Then, roll it out into an evenly thin sheet by using a glass rod on the glass plate. After drying at room temperature, the freestanding PTCDA electrodes are obtained by cutting the thin sheet into disks with a diameter of 12 mm and then drying in a vacuum oven at 60 °C for 8 h. The mass loading of the electrode is about 2.5–3 mg cm$^{-2}$.

**Preparation of Prussian blue (PB) electrode**. The preparation method of PB is based on the reported literature[55]. The detailed method is as follows: 0.4224 g $K_4Fe(CN)_6 \cdot 3H_2O$ is dissolved in 160 mL of deionized water and mixed well. At the same time, 0.3244 g $FeCl_3$ is dissolved in 160 mL of deionized water and mixed well. Then the $FeCl_3$ solution is added to the $K_4Fe(CN)_6$ solution drop by drop under stirring. The obtained mixture is then continued to be stirred for 2 h and aged for 24 h. The product is obtained by centrifugation and washed by extraction with deionized water and ethanol. The Prussian blue nanoparticles are eventually obtained after being vacuum dried at 80 °C for 24 h and are well consistent with the crystal structure of PB ($Fe_4(Fe(CN)_6)_3$ PDF#01-0239) as Supplementary Fig. S31. The PB electrode is prepared by mixing 60 wt% of active materials (PB), 30 wt% conductive agent (acetylene black), and 10 wt% binder (polyvinylidene fluoride) in the N-methyl pyrrolidinone. Then, the fully mixed slurry is uniformly pasted on carbon-coated aluminum foil and dried in a vacuum oven for 8 h at 80 °C. The mass loading of the active material is ~1.5 mg cm$^{-2}$.

**Electrochemical measurements**. CR2032 type coin cells are assembled in a standard argon-filled glove box. The separators are chosen as glass fiber (GF, for symmetric and full-cell electrochemical measurements) or polypropylene (PP, Celgard 2325) and GF with a configuration of PP/GF in cells for SEM, Raman and XPS characterizations (PP separator were used for nondestructive characterization since the direct contact between the fibrous and K electrodes would destroy the surface morphologies and chemical information of the surface of the K metal electrodes.). Two kinds of electrolyte were used: one is the highly concentrated electrolyte (3 M potassium bis(fluoroslufonyl)imide (KFSI) in EC/DEC = 1:1 VOL% (volume ratio = 1:1)) for symmetric battery system and SEM, Raman, XPS, EIS and Tafel plots measurements, the other is the KPF$_6$-based electrolyte (1 M potassium hexafluorophosphate (KPF$_6$) in EC/DEC (1:1, v/v)) for both symmetric battery system and full battery system. MSK-110 is used for cell assembly. CCPP, DNPP or pristine K electrodes were used as the working electrode and counter electrode for K metal symmetric cells. During the full battery system, the PTCDA and PB electrodes are used as the cathode. Every coin battery used about 60 μL (symmetric battery in KFSI-based electrolyte)/100 μL (symmetric battery in KPF$_6$-based electrolyte and full battery) electrolyte and all the charge-discharge tests are operated on a Neware

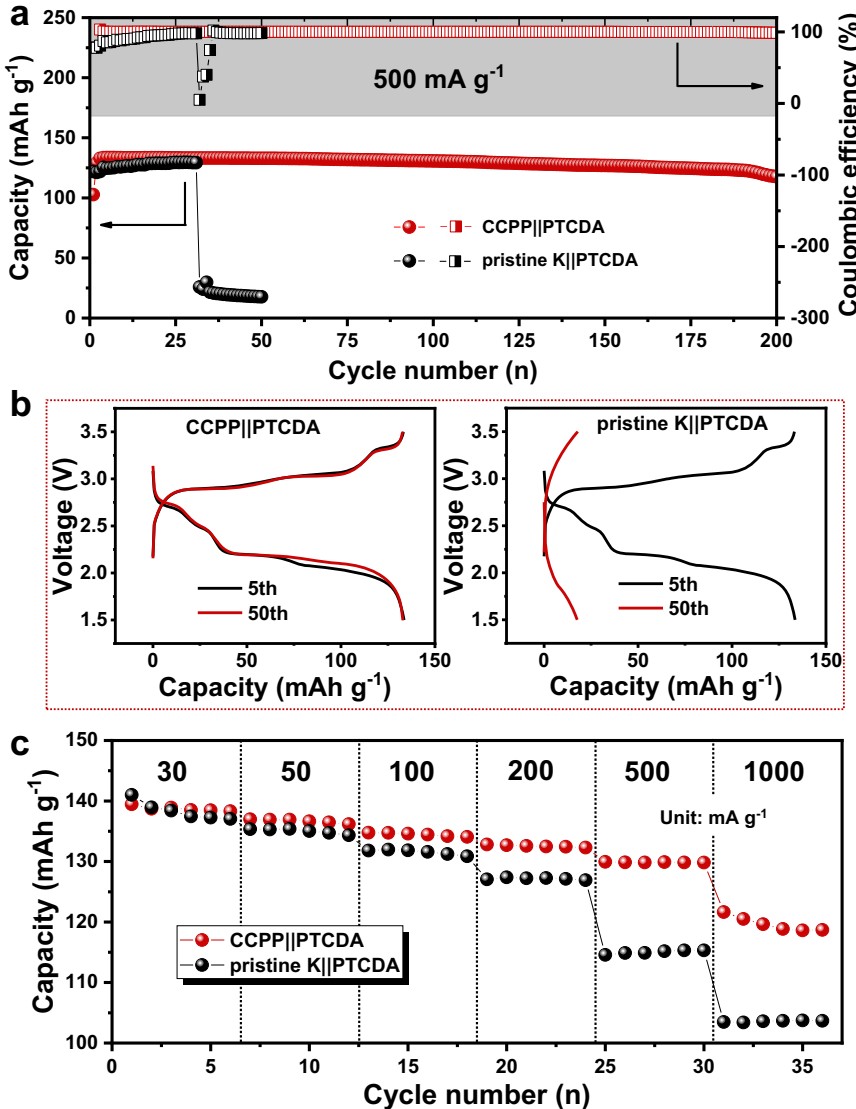

**Fig. 6 Full-cell performances in the KPF₆/EC:DEC electrolyte. a** Long-term stability of CCPP||PTCDA and pristine K||PTCDA batteries at 500 mA g⁻¹. **b** The 5th and 50th charge-discharge profiles of CCPP||PTCDA and pristine K||PTCDA at 500 mA g⁻¹. **c** The rate performances of CCPP||PTCDA and pristine K||PTCDA cells. Source data are provided as a Source Data file.

battery testing system with a constant temperature of 27 ± 0.1 °C (full cells) or room temperature (Symmetric cells).

Electrochemical impedance spectroscopy (EIS) and Tafel plots of cells are characterized by using the electrochemical workstations (IVIUM-VERTEX. C, Netherlands, and CHI760E, China). The frequency range of the EIS is from 100,000 to 0.1 Hz, as well as the disturbance amplitude was 0.005 V. Tafel plots and derived exchange current density obtained from linear sweep voltammetry tests in the range of −0.3 V to 0.3 V at 2 mV s⁻¹.

Electronic resistivity of SEI of CCPP is measured by a direct current–voltage curve on a Neware battery system. Herein, blocking electrodes with pristine K or CCPP sandwiched between two stainless steels are assembled. First, the blocking electrode is run at an open-circuit voltage for 20 s, then a constant current of 5 mA is applied for 80 s. The calculation formula is:

$$\rho = \frac{R \cdot S}{L} = \frac{U \cdot S}{I \cdot L} \tag{3}$$

Where $L$ is the thickness of SEI; $I$ is the applied current; $S$ is the contact area between the sample and stainless steel; $U$ is the average voltage increase.

**Materials characterizations**. Morphology images with EDS analyses are collected on Hitachi FE-SEM, S4800 at 5 KV and MIRA3 TESCAN (Hunan Navi New Materials Technology) at 10 KV with the exposure time to air below 20 s. The XRD patterns are recorded on a Bruker D8 Advance (Bruker AXS, Germany) with Cu Kα radiation (λ = 0.15418 nm). The sample is covered with kapton tape to ensure that the K-containing samples are not oxidized in the air during XRD

measurement. Raman spectra of samples are measured by using an Alpha300R Raman spectrometer system with a 532 nm laser as an excitation source. XPS measurements are carried out on a Thermo Scientific K-Alpha+ X-ray photo-electron spectrometer (Thermo Fisher). The mechanical tests are measured by a universal tensile testing machine (ZQ-990LB) with the extension rate at 20 mm min⁻¹. The pristine K and CCPP samples are cut into 6 × 28 × 0.3 mm slender strips and then wrapped and sealed into two pieces of 12 × 34 × 0.5 mm acrylic adhesive to ensure the samples are not changed in air, as shown in Supplementary Fig. S13. The preparation processes of the sample strips are always in the Ar gas-protected glove box.

**Charge test**. In order to output the charge generated during the tribo-electrochemistry reaction, a charge output device is used (Supplementary Fig. S5d). In this device, K metal, stainless steel mesh, and PTFE film or PTFE plate are stacked in layers (this work is carried out in a Ar-filled glove box), and the stainless steel mesh is grounded through a long wire. I–t curves is selected to test the output current, herein, the electrochemical workstation (CHI760E, China) is added and connected between the reaction device and the earth.

**Calculation of the amount of soluble K in defluorination reaction**. The amount of soluble K in the defluorination reaction is calculated by the relation between reaction heat and specific heat capacity, as shown in Eq. (4):

$$C_P = \frac{Q}{m\Delta T} \tag{4}$$

Where $Q$ is the released heat by the reaction; $C_P$ is the specific heat capacity of K, 0.76 KJ mol$^{-1}$ K$^{-1}$; $m$ is the mass of K that can be fused; $\Delta T$ is the temperature difference from a certain temperature to the melting point.

## Data availability

All data that support the findings of this study are available within the paper and its Supplementary Information or from the corresponding authors upon reasonable request. Source data are provided with this paper.

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

## Acknowledgements

Yingpeng Wu acknowledges financial support from National Natural Science Foundation of China (Grant No. 22075073, No. 21805079), Fundamental Research Funds for the Central Universities (531107051077). Lu Huang acknowledges financial support from National Natural Science Foundation of China (Grant No. 21805078), Fundamental Research Funds for the Central Universities (531107051042).

## Author contributions

Y.W. conceived the idea. C.Q. carried out the main synthesis, characterization, and electrochemical experiments. C.Q. and Y.L. repeated the main results. D.W., P.Y., T.X., H.Z., and G.L. participated in some of the experiments. Y.W., C.Q., D.W., P.Y., and L.H. analyzed the results. Y.W., L.H., and D.W. supervised the research. C.Q. and D.W.

drafted the manuscript. Y.W. and L.H. revised the manuscript. All authors participated in the interpretation of the data and production of the final manuscript.

## Competing interests

The authors declare no competing interests.
