## [Peer Review File · Nature Communications]

REVIEWER COMMENTS

Reviewer #1 (Remarks to the Author):

This manuscript reported a novel self-catalyzed piezochemistry process to construct a continuous and compact protective artificial SEI (ASEI) on K metal anode. The unique ASEI strategy solved the interface issues between “traditional” ASEI and electrode surface, so the cell performance in the work is impressive. Despite the requirement of further improvement, the manuscript is suggested for publication in Nature Commun. Detailed comments are as follows :

Comment 1: I recommend more discussion about the effects of CCPP anodes on potassium electrochemical plating/stripping behaviors. For example, the capacity and voltage platform are too low compared with other potassium metal or potassium graphite batteries in full cells (J. Mater. Chem. A 2017, 5, 4325-4330).

Comment 2: The sequence of Figure 1a, b, c should be switched because Figure 1b is firstly mentioned in paper instead of Figure 1a.

Comment 3: Figure S5d and Figure 1c seems to be right in page 5 (line 120, 121), not “Fig S5d” and “Figure 1C”.

Comment 4: Why is the C distribution different from F and K in Figure 2d-g?

Comment 5: White residual PTFE can be clearly observed on the CCPP anode, so how to ensure the consistency and repeatability of different CCPP anodes. Furthermore, the comparison of various CCPP anodes is not shown in the article.

Comment 6: The XRD patterns of K (PDF #01-0500) are different in Figure S11a and Figure S11b. How to explain this phenomenon?

Comment 7: The labels of Figure S28, S29 in page 15, 16 can not correspond to the Figure S28 and Figure S29 in the supplemental information.

Comment 8 : The Figure S30, Figure S31 and Figure S32 mentioned in page 15, 16 can't be found in supplemental information.

Comment 9 : The initial charge/discharge curves for CCPP/KPB (K/KPB) batteries should be provided in Figure 6 or in the supplemental information.

Comment 10: There are many other errors : i.e., the explanation of CCPP should be given for the first time in page 3 rather than in page 7; Journal abbreviation errors such as the references 18 and 23.

Reviewer #2 (Remarks to the Author):

The authors propose an interesting approach for creating a robust conducting SEI for potassium ion batteries. The authors argue that their artificial SEI effectively suppresses dendrites albeit the performance of their full battery is not on par or superior to those reported in the literature. Compared to the full batteries with ASEI (doi.org/10.1002/adv.202003639; ACS Nano 15, 9167–9175 (2021)) and smart anodes (Energy and Environmental Science, 14, 965-974 (2021); <https://doi.org/10.1093/nsr/nwaa276>; <https://doi.org/10.1021/acsmaterialslett.0c00171>) reported for potassium ion batteries, the capacity decay of the full battery in the present study is rapid even with the CCAM electrode in the full battery. In other words, their full battery didn't show any advantages than others reported in the literature.

This reviewer thinks that tribo- rather than piezo-electric effect is responsible for the creation of the KF-based robust artificial SEI because it is well known that PTFE is a strong electronegative material in the triboelectric series (see for example, Nano Energy 44 (2018) 103–110). Also, the authors used two methods - impacting the K anode (e.g., Fig. S7a) or rubbing over the K anode (e.g., Fig. S7b) in their study. Both methods relate to triboelectricity than piezoelectricity.

Recommended corrections:

In Fig. 2i, the Raman spectrum of PTFE must be expanded by 3x or 5x so that strongest Raman peak in PTFE at 734 cm^{-1} is easily visible. This peak is absent in the CCPP further confirming that CCPP is different from PTFE.

In Fig. S11, the XRD pattern of K PDF#0-0500 in panel (a) is different from the pattern of K PDF#0-0500 shown in panel (c). Why?

Minor corrections:

In Fig. S7, ...the illustration in (A) is an optical photograph... should be corrected to ...the illustration in (a) is an optical photograph...

This reviewer recommends publication after revision if the method can be refined further to elicit a better performance at the full cell because reports of superior ASEI for the full cell already exist in the literature.

Reviewer #3 (Remarks to the Author):

This manuscript presents a study of new artificial SEI (ASEI) strategy for K metal batteries (not for K-ion batteries). The ASEI derived from PTFE through piezochemistry reaction prevents short-circuit of K cells due to dendrite formation. The finding in this study provides an important contribution to long-life K-metal batteries and will possibly induce further development of this field. The referee, however, thinks that major revision and additional data are highly required for further round to be acceptable as follows.

1. If the electrolyte penetrates into the ASEI, this is not SEI, not passivation, but a protective layer and coating layer like a separator. In addition to K⁺ ion conductivity and electronic insulation, electrolyte penetration should also be examined. Considering the obtained results here, the title is misleading, and the referee suggests "PTFE-derived surface layer on K metal electrode" as an alternative title. The referee has a concern about the existence of SEI derived from electrolytes under the protective layer. If the ASEI passivates K metal, the type of electrolyte should not have a significant effect on the properties. However, in this study, highly concentrated KFSA/EC:DEC and KPF6/DME electrolytes, which form stable SEI on K metal, were used. The authors should add the results of symmetric and K/PBA cells filled with the most common KPF6/EC:DEC electrolyte to show that the electrochemical properties are independent of the electrolyte.

Although the following points are minor, they would be helpful for further improvement of the manuscript.

2. Figure 3c and Figure S16

The scales of the x and y axes of the Nyquist plot should be set equal.

3. K/PB cell should be charged to a higher voltage (~4.2 V) because PB might show additional plateau attributed to low spin Fe²⁺/³⁺ at ca 4.0 V.

4. (Experimental) The following can be a typo. The PTFE film (20 mm)

5. The approximate weight of the K metal electrodes should be mentioned because the capacity balance of K metal and PB is essential for the cycling performance of K/PB cells.

**The point-by-point response to the reviewers' comments.**

**Reviewer #1**

This manuscript reported a novel self-catalyzed piezochemistry process to construct a continuous
and compact protective artificial SEI (ASEI) on K metal anode. The unique ASEI strategy solved
the interface issues between “traditional” ASEI and electrode surface, so the cell performance in
the work is impressive. Despite the requirement of further improvement, the manuscript is
suggested for publication in Nature Commun.

**Response:** Thank you very much for your positive feedback. We have revised our manuscript
according to your helpful comments and valuable suggestions. Detailed responses are as follows:

**1、 Review comments:** I recommend more discussion about the effects of CCPP anodes on
potassium electrochemical plating/stripping behaviors. For example, the capacity and voltage
platform are too low compared with other potassium metal or potassium graphite batteries in full
cells (J. Mater. Chem. A 2017, 5, 4325-4330).

**Response:** Thank you very much for your valuable suggestion. We have added more discussions
about the effects of CCPP anode on potassium electrochemical plating/stripping behaviors,
including symmetric and full cells. The cathode (commercial Prussian blue powder, $C_{18}Fe_7N_{18}$,
99%) we used in the previous manuscript is different from that in J. Mater. Chem. A 2017, 5,
4325-4330 (K-MnHCFE and K-FeHCFE, cited as Ref. 6). Due to the inherent properties of the
commercial Prussian blue (PB) powder, it brings low capacity and voltage platform in the full
battery. However, compared with other reported potassium metal batteries with Prussian blue
analogue (PBA) as the cathode, the voltage platform within the voltage window of 0-4 V is not
low (Ref. Natl Sci Rev, 2020, 0, nwa276 and Ref. ACS Nano 2021, 15, 9167-9175). In order to
improve the capacity of full cells, we re-selected the cathode for full battery testing. Particularly,
the synthetic Prussian blue made in our lab could deliver an improved capacity of $>70 \text{ mAh g}^{-1}$
(see Figure S29), which is consistent with the capacity and voltage platform in the Ref (ACS
Nano 2021, 15, 9167-9175). The followings are the detailed discussion and corresponding
changes:

(1) For symmetric cells, we have added new data by using 0.8 M KPF₆/EC:DEC electrolyte
in the manuscript (MS) and supporting information (SI).

In the MS, page 14, line 308-321, we added the discussions: In addition, the CCPP anodes
also exhibits remarkable superiority in KPF₆/EC:DEC electrolyte. Under a large current density
of 2 mA cm⁻², maintaining a stable polarization for more than 110 h (Figure 4c). In contrast,
DNPP and pristine K become short circuit after 56 h and 35 h respectively, due to the lack of
stable and robust SEI protection. Figure 4d shows the voltage profiles at 0.5 mA cm⁻² and 0.5
mAh cm⁻², pristine K and DNPP short circuit ascribe to the dendrites within 90 h and 140 h,
respectively. CCPP, by contrast, can run stably more than 250 h, which is almost 3 times that of
pristine K. For the rate performance, CCPP can withstand circulation at higher current densities
(0.1 mA cm⁻² to 0.5, 1, 1.5, 2 and 3 mA cm⁻²). Whereas, pristine K cannot last more than 4
cycles at 1.5 mA cm⁻² and 1 mAh cm⁻² (Figure S23). In a word, CCPP possess better
performances not only in the highly concentrated KFSI/EC:DEC electrolyte but also in
KPF₆/EC:DEC electrolyte, which indicate that the importance of a continuous and compact
robust SEI to inhibit K metal dendrites.

In the MS and SI, we have added the following Figures:

Figure 4. (c) Polarizations of CCPP, DNPP and pristine K at 2 mA cm⁻² and 0.5 mAh cm⁻² by
using KPF₆/EC:DEC electrolyte.

Figure 4. (d) Polarizations of CCPP, DNPP and pristine K at 0.5 mA cm⁻² and 0.5 mAh cm⁻² by
using KPF₆/EC:DEC electrolyte.

Figure S23. Rate performances of CCPP and pristine K at different current densities in the
 KPF₆/EC:DEC electrolyte.

(2) For full cells, we used two kinds of cathode: commercial
 Perylene-3,4,9,10-tetracarboxylic dianhydride (PTCDA, the mass loading of PTCDA is about
 2.5-3 mg cm⁻²), synthetic PB (the mass loading of PB is about 1.5 mg cm⁻²) cathode. The
 following discussions were also added correspondingly:

In the manuscript, page 16, line 365-383, we added: Tests of full cells matched with
 PTCDA confirmed the feasibility of CCPP anodes in real applications. Figure 6a shows the
 long-term stability of CCPP||PTCDA and pristine K||PTCDA at the current density of 500 mA
 g⁻¹, the CCPP||PTCDA cell shows a discharge capacity of 133.56 mAh g⁻¹ (5st cycle) when
 stabilized for a number of cycles and keep a capacity of 117.5 mAh g⁻¹ after 200 cycles with a
 capacity retention of 88% (Figure 6b). However, pristine K||PTCDA is broken soon after 31
 cycles because of the intensified dendrites growth and electrolyte decomposition for the lack of
 stable SEI protection. Furthermore, the CCPP||PTCDA cell delivers reversible capacities of
 138.8, 136.8, 134.6, 132.6, 130, 120 mAh g⁻¹ at current density of 30, 50, 100, 200, 500, 1000
 93 mA g⁻¹, respectively (Figure 6c and Figure S28). By contrast, pristine K||PTCDA cell deliver
 138.4, 135, 131.5, 127.2, 115, 103.6 mAh g⁻¹, respectively. These results direct to superb rate
 capability of CCPP||PTCDA cell. In addition, CCPP||PB and pristine K||PB cells were also
 assembled and tested to verify the effects of the continuous and compact SEI. As shown in
 Figure S29, although both cells deliver comparable initial capacity of about 77 mAh g⁻¹, the
 capacity of pristine K||PB drops to 67.7 mAh g⁻¹ after 150 cycles (Figure S29c). The CCPP||PB
 cell, in contrast, could deliver a higher capacity of 74.9 mAh g⁻¹ (Figure S29b), indicating
 reduced loss of electrolyte and capacity of CCPP||PB cell.

In the MS and SI, we have added the following Figures:

**Figure 6. (a)** Long-term stability of CCPP||PTCDA and pristine K||PTCDA batteries at 500 mA
g⁻¹.

**Figure 6. (b)** The 5th and 50th charge-discharge profiles of CCPP||PTCDA and pristine
K||PTCDA at 500 mA g⁻¹.

**Figure 6. (c)** The rate performances of CCPP||PTCDA and pristine K||PTCDA cells.

**Figure S28.** Charge and discharge profiles of (a) CCPP||PTCDA and (b) K||PTCDA at different

current rates.

**Figure S29.** (a) Long-term cycling performance of CCPP||PB and pristine K||PB cells at 100 mA

g^{-1} by using KPF₆/EC:DEC electrolyte. (b) The 1st and 150th charge-discharge profiles of

CCPP||PB cells at 100 mA g^{-1} . (c) The 1st and 150th charge-discharge profiles of pristine K||PB

cells at 100 mA g^{-1} .

**2、Review comments:** The sequence of Figure 1a, b, c should be switched because Figure 1b is
firstly mentioned in paper instead of Figure 1a.

**Response:** Thank you for your valuable suggestion. We have switched the layout of Figure 1 and
renewed the corresponding legend and label in the manuscript according to your reasonable
advises.

In the manuscript, Page 4, the Figure 1a, b, c were switched. At the same time, we also
changed the legend in Figure 1 a, b, c accordingly.

**Figure 1. Self-catalysis Tribo-electrochemistry Mechanism as well as Related**
**Experiments. (a) Schematic diagram of explosive reaction and K liquefaction when strong**
**pressure is applied to the joint surface between PTFE and K metal. (b) Schematic diagram**
**of self-catalysis tribo-electrochemistry reaction process. (c) The output current with regular**
**force applied to the grounded joint surface between PTFE and K metal.**

**3、 Review comments:** Figure S5d and Figure 1c seems to be right in page 5 (line 120, 121), not
“Fig S5d” and “Figure 1C”.

**Response:** Thank you for your valuable suggestion. In the MS, page 5, line 124, 125, we
modified “Fig S5d” and “Figure 1C” to “**Figure S5d**” and “**Figure 1c**”.

**4、 Review comments:** Why is the C distribution different from F and K in Figure 2d-g?

**Response:** Thank you very much for highlighting this issue. The distribution direction and trend
of C, F and K are in fact consistent. The C distribution “seems” different from F and K, which
may be originated from valid pixels of C collected by the instrument are far less than those of F
and K. The difference of the collected effective pixels essentially comes from the difference of
the characteristic X-ray intensities of the three.

**Different elements possess different signal intensity by characteristic X-rays: Energy**
**dispersive spectrometer (EDS)** is mainly used for qualitative analysis of elements by measuring
the energy of the characteristic X-ray excited in the material and for quantitative analysis of
elements by measuring the intensity of the characteristic X-rays. When EDS works, the
characteristic X-rays excited from the sample directly irradiate the semiconductor detector to
generate electron-hole pairs, the number of which is proportional to the X-ray energy, namely:

$$N=E/\varepsilon$$

Where ε is the energy (3.8 eV) produced by generating an electron-hole pair, and the E is
the photon energy of the characteristic X-ray.

Then electron-hole pairs can be separated and collected, converted into output pulse height
which is determined by N, forming the abscissa of the EDS map: energy. According to the
number of characteristic X-rays recorded in different intensity ranges, the intensity of X-rays of
different elements can be determined, forming the ordinate of the EDS map: intensity (Ref.
Nuclear Instruments & Methods, 1977, 142(1-2), 213-223). There are two main reasons for the
lower intensity of C element. Firstly, the energy of the characteristic X-rays emitted by different
elements is different. For C element, the photon energy of the characteristic X-ray is only 277 eV
($K\alpha$), while F element is 677 eV ($K\alpha$) and K element is 3310 eV ($K\alpha$). Such low photon energy
of C element can be easily absorbed by the matrix, causing lower intensity and less pixel in the
mapping results. Moreover, the characteristic X-ray fluorescence yield of different elements is

different based on the equation (Ref. Reviews of Modern Physics, 1966, 38(3): 513-540):

$$\omega_k=(1+a/Z^4)^{-1}$$

Where a is a constant (when the atomic number lower than 30) and Z is the atomic number.

For C element, the ω_k is only ~ 0.28% while F is 1.3% and K is 14%. (Ref. X-Ray Spec,
1986, 15(4): 271-285, J. Phys. Chem. Ref. Data, 1979, 8(2): 307-328, J. Phys. Chem. Ref. Data,
1994, 23(1): 339-364.)

Based on above, the intensity of C element is quite lower than F and K and their distribution
“seems” different. To avoid the signal noise and low signal strength, we used signal processing
technology to make sure the C, F and K have the same distribution.

**Effective pixel statistics:** Through statistics (Code operation, see Figure R1), we found that the
lowest number of valid pixel points is C (42196), compare to F (150296) and K (118429).

**The distribution direction and trend of the three are consistent:** Through further Fourier
transform (FFT) (by using Image Analyzer-version 1.34, MeeSoft) of the mapping of C, F, K
element, it can be clearly seen that the distribution directions and trends of the three are
consistent (see Figure R2). The FFT function converts information present in an original data
image from “real” space into mathematically defined “frequency” space (Ref. Neuron Glia
Biology, 2006, 2(02), 93-103). The FFT of an image containing aligned structure results in an
output image containing pixels distributed in a non-random, flat aperture distribution (Ref.
Biomaterials, 2006, 27 (32), 5524-5534). This distribution occurs because the pixel intensities
are preferentially distributed with a specific orientation (Ref. J Biomater Sci Polymer Ed, 2008,
19 (5): 603-621). As shown in Fig R2, the distribution directions and trends of the three elements
are consistent.

Codes:

```
import cv2 as cv
# Read image pixel data of C.tiff, F.tiff, K.tiff
C_data = cv.imread(r"F:\qcc\C.tiff", -1)
F_data = cv.imread(r"F:\qcc\F.tiff", -1)
K_data = cv.imread(r"F:\qcc\K.tiff", -1)
x, y, z = C_data.shape
cnt = [0, 0, 0] # Save the number of valid C,F,K pixels counted
# Calculating valid pixel points in C.tiff
c_data = C_data.reshape(x * y, z)
# Each pixel contains a RGB three-channel data to represent the colour,
# from left to right, blue, green and red, i.e. x[0] is the blue value,
# x[1] is the green value and x[2] is the red value, x represents a pixel
for c in c_data:
# Conditions for a valid pixel point judged to be C: c[2](red) channel pixel value
# is greater than 0, and c[1](green), c[2](blue) values are both 0
if (c[0] == 0 and c[1] == 0) and c[2] > 0:
cnt[0] += 1 # cnt[0] represents the number of valid pixels in C.tiff
# Calculating valid pixel points in F.tiff
f_data = F_data.reshape(x * y, z)
for f in f_data:
# Conditions for a valid pixel to be judged as F: f[2](red) is equal to 0, and f[0](blue)
# and f[1](green) are greater than 0, and the blue value is equal to the green value
if (f[2] == 0) and (f[0] > 0 and f[1] > 0) and (f[0] == f[1]):
cnt[1] += 1 # cnt[1] represents the number of valid pixels in F.tiff
# Calculating valid pixel points in K.tiff
k_data = K_data.reshape(x * y, z)
for k in k_data:
# Conditions for a valid pixel to be judged as F: k[1] (green) value equal to 0 and
# k[0] (blue) and k[2] (red) greater than 0 and equal values of blue and red
if (k[1] == 0) and (k[0] > 0 and k[2] > 0) and (k[0] == k[2]):
cnt[2] += 1 # cnt[2] represents the number of valid pixels in K.tiff
print("the number of valid pixels in C.tiff is : {}".format(cnt[0]))
print("the number of valid pixels in F.tiff is : {}".format(cnt[1]))
print("the number of valid pixels in K.tiff is : {}".format(cnt[2]))
print("The highest number of valid pixel points is F ({}), followed by K ({}), "
"and the lowest is C({})".format(F_cnt=cnt[1], K_cnt=cnt[2], C_cnt=cnt[0]))
```

Results:

```
the number of valid pixels in C.tiff is : 42196
the number of valid pixels in F.tiff is : 150296
the number of valid pixels in K.tiff is : 118429
The highest number of valid pixel points is F (150296),
followed by K (118429) and the lowest is C(42196)
```

Figure R1. Codes and results for pixel point calculation.

Figure R2. The Fourier transform results of the mapping map of C, F and K element.

**5、 Review comments:** White residual PTFE can be clearly observed on the CCPP anode, so how
to ensure the consistency and repeatability of different CCPP anodes. Furthermore, the
comparison of various CCPP anodes is not shown in the article.

**Response:** Thank you very much for noting this.

**5-a:** In order to ensure the consistency and repeatability of different CCPP anodes, in our
experiment, the CCPP anodes were prepared by artificial repeatedly applying friction or pressure
on the joint surface until an intact black film appeared on the K foil surface, after that, peeling
off the top layer of unreacted PTFE film, detailed steps are as shown in Figure S7b. Due to the
tight adhesion between the product layer and the main body, some white PTFE remains on the
surface of electrode uncontrollably. In our work, we mainly choose samples with a complete
black protective layer for electrochemical testing.

Herein, we proved the repeatability of prepared samples by structural characterization,
component identification and impedance analysis of different samples.

Firstly, SEM and EDS characterization (Figure R3) shows that different samples have
similar morphology and close content of C, F, K element.

Figure R3. SEMs and EDSs of three different samples.

Secondly, The XRD spectra reveal that all the components of samples are approaching an
 amorphous state (see Figure R4).

Figure R4. XRD patterns of three different CCPP samples.

Thirdly, the small difference in the resistance of symmetrical cells also confirms the good
 repeatability of different samples, which indicates that different samples have similar interface
 properties (see Figure R5).

Figure R5. EIS plots of three different CCPP samples by using $KPF_6/EC:DEC$ electrolyte.

**5-b:** In order to further prove the repeatability of different CCPP anodes in electrochemical
behavior, we give several sets of repeated dates of CCPP samples in symmetric and full cells.

Firstly, the long-term stability of CCPP symmetric cells at 0.5 mA cm^{-2} and capacity of 0.5
mAh cm^{-2} (see Figure R6), it can be seen that all of them can cycle steadily for long time.

Figure R6. Voltage-time profiles of CCPP at the current density of 0.5 mA cm^{-2} and capacity of
0.5 mAh cm^{-2} in the highly concentrated KFSI/EC:DEC electrolyte. (a) is the same as Figure
4a in MS, (b, c) are the same as Figure S18a, b in SI.

Secondly, full cell of CCPP samples at 500 mA g^{-1} for K||PTCDA batteries (see Figure R7).

Figure R7. Long-term stability of the CCPP symmetric cells at 500 mA g^{-1} by using
KPF₆/EC:DEC electrolyte.

**5-c:** Although there is a small amount of PTFE residue on the surface of the CCPP, the
PTFE residue in CCPP is negligible comparing with pure PTFE, which can be seen from the
XRD results (Figure S11c). Such a small amount of PTFE residue has a weak effect on the
consistency and repeatability of different CCPP anodes.

Figure S11. (c) XRD contrast patterns of pristine K, PTFE and CCPP in the same intensity range.

**6. Review comments:** The XRD patterns of K (PDF #01-0500) are different in Figure S11a and
Figure S11b. How to explain this phenomenon?

**Response:** Thank you so much for highlighting this issue and we apologize for this mistake.

We especially looked at the original vector drawings of Figure S11a and Figure S11b, and
found that the raw data of the two were consistent (as shown in Figure R8). In the last version,
the author originally wanted to adjust the size of lines to 3 when drawing, but accidentally adjusted
the line at $2\theta=23.707^\circ$ to 0, resulting in the loss of crystal form at $2\theta=23.707^\circ$.

To this end, we corrected the XRD patterns of K (PDF #01-0500) in Figure S11a.

**Figure R8.** The original vector drawing and raw data of Figure S11a and Figure S11b.

In the SI, page 7, we corrected the XRD patterns of K (PDF #01-0500) in Figure S11a.

**Figure S11.** (a) XRD pattern of pristine K.

**7、 Review comments:** The labels of Figure S28, S29 in page 15, 16 cannot correspond to the
Figure S28 and Figure S29 in the supplemental information.

**Response:** Thank you so much for reminding me of this issue and we apologize for this mistake.

We carefully proofread the labels in the MS and the Figures in the SI, and made one-to-one
correspondence. Due to new test of full cells, we removed previous discussion about full cell and
removed previous Figure S28 and Figure S29 correspondingly.

**8、 Review comments:** The Figure S30, Figure S31 and Figure S32 mentioned in page 15, 16
can't be found in supplemental information.

**Response:** Thank you so much for reminding me of this issue.

We carefully proofread the labels in the MS and the Figures in the SI, and made one-to-one
correspondence. Due to new test of full cells, we removed previous discussion about full cell and
removed previous Figure S30, Figure S31 and Figure S32 correspondingly.

**9、 Review comments:** The initial charge/discharge curves for CCPP/KPB (K/KPB) batteries
should be provided in Figure 6 or in the supplemental information.

**Response:** Thank you very much for your valuable suggestion. Due to the inherent properties of
the commercial PB, it brings low capacity and voltage platform in the full battery. Based on the
suggestion from reviewer 2, we replaced the previous commercial PB with synthetic PB made in
our lab. The corresponding initial charge/discharge curves were shown in Figure S29b, c (In the
SI, page 19).

Figure S29. (b) The 1st and 150th charge-discharge profiles of CCPP||PB cells at 100 mA g⁻¹. (c)

The 1st and 150th charge-discharge profiles of pristine K||PB cells at 100 mA g⁻¹.

**10、 Review comments:** There are many other errors: i.e., the explanation of CCPP should be
given for the first time in page 3 rather than in page 7; Journal abbreviation errors such as the
references 18 and 23.

**Response:** We are really grateful for your helpful reminders and apologize for these errors. We
examined the manuscript carefully for errors and highlighted the changes made in the manuscript
in red. For example:

(1) We first give the explanation of CCPP in page 3:

In the MS, page 3, line 73-line 74, we added “**Based on this phenomenon, continuous and**
**compact protected potassium (CCPP) anodes were prepared**”.

(2) We corrected the journal abbreviations in the references.

In the MS, page 22, line 547-line 548, the Journal abbreviation of the previous references 18
was corrected to “*J. Am. Chem. Soc.*”.

In the MS, page 23, line 558, the Journal abbreviation of the previous references 23 was
corrected to “*Nat. Commun.*”.

**Reviewer #2**

The authors propose an interesting approach for creating a robust conducting SEI for potassium
ion batteries. The authors argue that their artificial SEI effectively suppresses dendrites albeit the
performance of their full battery is not on par or superior to those reported in the literature.
Compared to the full batteries with ASEI (doi.org/10.1002/adv.202003639; ACS Nano 15,
9167–9175 (2021)) and smart anodes (Energy and Environmental Science, 14, 965-974 (2021);
https://doi.org/10.1093/nsr/nwaa276; https://doi.org/10.1021/acsmaterialslett.0c00171) reported
for potassium ion batteries, the capacity decay of the full battery in the present study is rapid
even with the CCAM electrode in the full battery. In other words, their full battery didn't show
any advantages than others reported in the literature.

**Response:** Thanks a lot for your useful comments. The experiments show that the low capacity
and rapid capacity decay are mainly determined by the commercial PB cathode rather than the
anodes. According to your suggestion, we first try to optimize the performance by using the
previous commercial Prussian blue (PB) as the cathode. Although CCPP||PB has significant
advantage over pristine K||PB, it still cannot increase its capacity (Figure R9a and Figure R9b).

**Figure R9.** (a) Long-term stability of CCPP||commercial PB and pristine K||commercial PB
batteries at 100 mA g⁻¹ in the highly concentrated KFSI/EC:DEC electrolyte, the mass loading of
PB is about 4 mg cm⁻². (b) Long-term stability of CCPP||commercial PB and pristine
K||commercial PB batteries at 100 mA g⁻¹ in KPF₆/EC:DEC electrolyte, the mass loading of PB
is about 1.5 mg cm⁻².

Based on this, we re-matched two kinds of cathode: commercial
Perylene-3,4,9,10-tetracarboxylic dianhydride (PTCDA, the mass loading of PTCDA is about
2.5-3 mg cm⁻², see Figure 6a, d, e), synthetic Prussian blue made in our lab (PB, the mass loading

of PB is about 1.5 mg cm^{-2} , see Figure S29a) cathode. The performances are greatly increased
and the new results are added in manuscript (MS) and supporting information (SI):

In the MS, page 16, line 365-383, we added: Tests of full cells matched with PTCDA
confirmed the feasibility of CCPP anodes in real applications. Figure 6a shows the long-term
stability of CCPP||PTCDA and pristine K||PTCDA at the current density of 500 mA g^{-1} , the
CCPP||PTCDA cell shows a discharge capacity of $133.56 \text{ mAh g}^{-1}$ (5^{st} cycle) when stabilized for
a number of cycles and keep a capacity of 117.5 mAh g^{-1} after 200 cycles with a capacity
retention of 88% (Figure 6b). However, pristine K||PTCDA is broken soon after 31 cycles
because of the intensified dendrites growth and electrolyte decomposition for the lack of stable
SEI protection. Furthermore, the CCPP||PTCDA cell delivers reversible capacities of 138.8,
136.8, 134.6, 132.6, 130, 120 mAh g^{-1} at current density of 30, 50, 100, 200, 500, 1000 mA g^{-1} ,
respectively (Figure 6c and Figure S28). By contrast, pristine K||PTCDA cell deliver 138.4, 135,
131.5, 127.2, 115, 103.6 mAh g^{-1} , respectively. These results direct to superb rate capability of
CCPP||PTCDA cell. In addition, CCPP||PB and pristine K||PB cells were also assembled and
tested to verify the effects of the continuous and compact SEI. As shown in Figure S29, although
both cells deliver comparable initial capacity of about 77 mAh g^{-1} , the capacity of pristine K||PB
drops to 67.7 mAh g^{-1} after 150 cycles (Figure S29c). The CCPP||PB cell, in contrast, could
deliver a higher capacity of 74.9 mAh g^{-1} (Figure S29b), indicating reduced loss of electrolyte
and capacity of CCPP||PB cell.

In the MS and SI, we have added the following figures:

**Figure 6.** (a) Long-term stability of CCPP||PTCDA and pristine K||PTCDA batteries at 500 mA
g^{-1} .

**Figure 6. (c)** The rate performances of CCPP||PTCDA and pristine K||PTCDA cells.

**Figure S29. (a)** Long-term cycling performance of CCPP||PB and pristine K||PB cells at 100 mA
 g⁻¹ by using KPF₆/EC:DEC electrolyte.

In the MS, we have cited the following references: line 443, we have cited Ref. ACS Nano
 15, 9167–9175 (2021) as Ref 52; line 193, we have cited Ref. Energy and Environmental
 Science, 14, 965-974 (2021) as Ref 32; line 36, we have cited Ref.
 doi.org/10.1002/advs.202003639 as Ref 9.

**1、 Review comments:** This reviewer thinks that tribo- rather than piezo-effect is responsible for
 the creation of the KF-based robust artificial SEI because it is well known that PTFE is a strong
 electronegative material in the triboelectric series (see for example, Nano Energy 44 (2018) 103–
 110). Also, the authors used two methods - impacting the K anode (e.g., Fig. S7a) or rubbing
 over the K anode (e.g., Fig. S7b) in their study. Both methods relate to triboelectricity than
 piezoelectricity.

**Response:** We are really grateful for this insightful comment, which will significantly improve
 the quality of our manuscript. According to the suggestion and literature research, we believe that
 tribo- rather than piezo-effect is responsible for the creation of the KF-based robust artificial SEI,

and reasons are as follows:

(1) From the definition of tribo- and piezo-effect, the construction method of SEI is more
consistent with tribo-effect.

Tribo-effect: A contact-induced electrification in which a material becomes electrically
charged after it is contacted with different material through friction. After two different materials
come into contact, a chemical bond is formed between some parts of the two surfaces, called
adhesion, and charges move from one material to the other to equalize their electrochemical
potential, When separated, some of the bonded atoms have a tendency to keep extra electrons
and some a tendency to give them away, possibly producing triboelectric charges on surfaces
(Ref. ACS Nano, 2013, 7 (11): 9533.). Herein, relative displacement is required between two
friction layers (Nano Energy, 38 (2017) 43–50).

Piezo-effect: Some dielectrics will deform when subjected to external forces in a certain
direction, resulting in internal polarization and the simultaneous appearance of positive and
negative charges on its two opposite surfaces (Ref. Science, 2006, 312 (5771): 242-246.).
Typical piezoelectric materials including ZnO NWs (Ref. Science, 2006, 312 (5771):
242-246.), lead zirconate titanate (PZT) (Ref. Adv. Mater., 2014, 26: 2514–2520), prestigious
PVDF and its copolymers (Ref. Adv. Mater., 2015, 27: 2340–2347).

(2) As the reviewer said, PTFE is a strong electronegative material in the triboelectric series,
which has been widely used in TENGs for achieving large potential differences (Ref. Nano
Energy 44 (2018) 103–110). In the MS, line 68, we have cited the Ref. Nano Energy 44 (2018)
103–110 as Ref 20.

Based on it, we have changed the title and the related statements and figures of the MS.

In the MS, page 6, line 130–line 136, we discussed the mechanism in detail: Taking these
into consideration, we can conclude that the powerful reaction between solid K and PTFE can be
triggered at room temperature when the negative charges accumulate on PTFE. Once a pressure
is applied, a lateral shear occurs between K and PTFE (Ref. ACS Nano 7 (2013) 9533-9557).
That is, there is a friction interaction between the two, causing the transfer of charge between
each other, as shown in Figure 2b (□). The extra electrons lead to a high tribo-electric field
(Figure 2b (□)), acting as the catalyzer and triggering the reaction. And external force is only the
initiator to the charge.

We made corresponding revision to Figure 2b (□).

Meanwhile, we renamed the title “Tribo-electrochemistry induced Artificial SEI by

Self-catalysis”, and all the term “piezo-” in this manuscript have been corrected.

In the MS, page 4, we updated Figure 1:

Figure 1. Self-catalysis Tribo-electrochemistry Mechanism as well as Related

Experiments. (a) Schematic diagram of explosive reaction and K liquefaction when strong

pressure is applied to the joint surface between PTFE and K metal. (b) Schematic diagram of

self-catalysis tribo-electrochemistry reaction process. (c) The output current with regular force

applied to the grounded joint surface between PTFE and K metal.

**2. Review comments:** In Fig. 2i, the Raman spectrum of PTFE must be expanded by 3x or 5x so
that strongest Raman peak in PTFE at 734 cm^{-1} is easily visible. This peak is absent in the CCPP
further confirming that CCPP is different from PTFE.

**Response:** We appreciate your valuable suggestion, and the expanded Raman spectrum of PTFE
was presented in Figure S12a. Meanwhile, we expanded the Raman spectrum of PTFE in Figure
2i. It is clear that there is indeed a peak in PTFE at around 734 cm^{-1} (represent CF_2 symmetric
stretching, Ref. ACS Sustainable Chem. Eng. 2019, 7, 17554-17558), however, it is not found on
CCPP, which means an adequate defluorination reaction occurs between K and PTFE.

According to your suggestion, in the MS, page 8, line 194-line 197, we added “Meanwhile,
a strongest Raman peak at 734 cm^{-1} (represent CF_2 symmetric stretching, Ref. ACS Sustainable
Chem. Eng. 2019, 7, 17554-17558) is easily visible in PTFE, but absent in the CCPP, which
further confirming that an adequate defluorination reaction occurs between K and PTFE.”.

In the MS, page 8, we redraw the Figure 2i by expanded the Raman spectrum of PTFE:

Figure 2. (i) Raman spectra of CCPP, pristine K and Pure PTFE.

In the SI, page 8, we presented Figure S12a:

Figure S12. (a) Raman spectrum of Pure PTFE.

**3、 Review comments:** In Fig. S11, the XRD pattern of K PDF#0-0500 in panel (a) is different
from the pattern of K PDF#0-0500 shown in panel (c). Why?

**Response:** Thank you so much for highlighting this issue and we apologize for this mistake.

We especially looked at the original vector drawings of Figure S11a and Figure S11b, and
found that the raw date of the two were consistent (as shown in Figure R8). In the last version,
the author originally wanted to adjust the size of lines to 3 when drawing, but accidentally adjusted
the line at $2\theta=23.707^\circ$ to 0, resulting in the loss of crystal form at $2\theta=23.707^\circ$.

To this end, we corrected the XRD patterns of K (PDF #01-0500) in Figure S11a.

**Figure R8.** The original vector drawing and raw data of Figure S11a and Figure S11b.

In the SI, page 7, we corrected the XRD patterns of K (PDF #01-0500) in Figure S11a.

Figure S11. (a) XRD pattern of pristine K.

**4、 Review comments:** In Fig. S7,the illustration in (A) is an optical photograph... should be
corrected to ...the illustration in (a) is an optical photograph...

**Response:** Thank you very much for your valuable suggestion. In the SI, page 4, line 36, we
corrected “....the illustration in (A) is an optical photograph...” to “...the illustration in (a) is an
optical photograph...”.

**5、 Review comments:** This reviewer recommends publication after revision if the method can be
refined further to elicit a better performance at the full cell because reports of superior ASEI for
the full cell already exist in the literature.

**Response:** Thanks for the recommends. According to the suggestion of this reviewer, we
optimized the performance of full cell by rematching commercial PTCDA (the mass loading of
PTCDA is about 2.5-3 mg cm⁻²) and synthetic Prussian blue (PB, the mass loading of PB is about
1.5 mg cm⁻²) cathode, respectively. The long-term stability and rate performances of K||PTCDA
full cells are given in Figure 6, meanwhile the long-term stability of K||PB full cells also have
been given in Figure S29. We also added some sentences in MS and SI.

In the MS, page 16, line 365-383, we have added: Tests of full cells matched with PTCDA
confirmed the feasibility of CCPA anodes in real applications. Figure 6a shows the long-term
stability of CCPA||PTCDA and pristine K||PTCDA at the current density of 500 mA g⁻¹, the
CCPA||PTCDA cell shows a discharge capacity of 133.56 mAh g⁻¹ (5st cycle) when stabilized for
a number of cycles and keep a capacity of 117.5 mAh g⁻¹ after 200 cycles with a capacity
retention of 88% (Figure 6b). However, pristine K||PTCDA is broken soon after 31 cycles
because of the intensified dendrites growth and electrolyte decomposition for the lack of stable
SEI protection. Furthermore, the CCPA||PTCDA cell delivers reversible capacities of 138.8,
136.8, 134.6, 132.6, 130, 120 mAh g⁻¹ at current density of 30, 50, 100, 200, 500, 1000 mA g⁻¹,
respectively (Figure 6c and Figure S28). By contrast, pristine K||PTCDA cell delivers 138.4, 135,
131.5, 127.2, 115, 103.6 mAh g⁻¹, respectively. These results direct to superb rate capability of
CCPA||PTCDA cell. In addition, CCPA||PB and pristine K||PB cells were also assembled and
tested to verify the effects of the continuous and compact SEI. As shown in Figure S29, although
both cells deliver comparable initial capacity of about 77 mAh g⁻¹, the capacity of pristine K||PB
drops to 67.7 mAh g⁻¹ after 150 cycles (Figure S29c). The CCPA||PB cell, in contrast, could

deliver a higher capacity of 74.9 mAh g^{-1} (Figure S29b), indicating reduced loss of electrolyte
and capacity of CCPP||PB cell.

In the MS and SI, we have added the following figures:

**Figure 6. (a)** Long-term stability of CCPP||PTCDA and pristine K||PTCDA batteries at 500 mA g^{-1}
g^{-1} .

**Figure 6. (c)** The rate performances of CCPP||PTCDA and pristine K||PTCDA cells.

**Figure S29. (a)** Long-term cycling performance of CCPP||PB and pristine K||PB cells at 100 mA g^{-1}
g^{-1} by using KPF₆/EC:DEC electrolyte.

**Reviewer #3**

This manuscript presents a study of new artificial SEI (ASEI) strategy for K metal batteries (not
for K-ion batteries). The ASEI derived from PTFE through piezochemistry reaction prevents
short-circuit of K cells due to dendrite formation. The finding in this study provides an important
contribution to long-life K-metal batteries and will possibly induce further development of this
field. The referee, however, thinks that major revision and additional data are highly required for
further round to be acceptable as follows.

**Response:** We are grateful for such a valuable comment and objective evaluation. According to
your suggestions, we have made major revisions and corrections to the whole manuscript.

**1、Review comments:** If the electrolyte penetrates into the ASEI, this is not SEI, not passivation,
but a protective layer and coating layer like a separator. In addition to K^+ ion conductivity and
electronic insulation, electrolyte penetration should also be examined. Considering the obtained
results here, the title is misleading, and the referee suggests “PTFE-derived surface layer on K
metal electrode” as an alternative title. The referee has a concern about the existence of SEI
derived from electrolytes under the protective layer. If the ASEI passivates K metal, the type of
electrolyte should not have a significant effect on the properties. However, in this study, highly
concentrated KFS/EC:DEC and KPF₆/DME electrolytes, which form stable SEI on K metal,
were used. The authors should add the results of symmetric and K/PBA cells filled with the most
common KPF₆/EC:DEC electrolyte to show that the electrochemical properties are independent
of the electrolyte.

**Response:** We are really grateful for this insightful and exploratory comment. We have made
further experiments and based on those results, we consider the KF-based robust layer of CCPP
to be “ASEI”. The results conform to the main characteristic of “ASEI” and possess better
performances not only in the highly concentrated KFS/EC:DEC electrolyte but also in
KPF₆/EC:DEC electrolyte. Here are the discussions in details.

**1-a:** Reviewer comments “In addition to K^+ ion conductivity and electronic insulation,
electrolyte penetration should also be examined.”

**Response to 1-a: (1)** 1979, Peled firstly realized the electrically insulating and ionically
conductive interface and named it as the solid electrolyte interphase (SEI) (Ref. E. Peled , J.

Electrochem. Soc. 1979, 126, 2047 .). We have proved CCPP improved K^+ ion conductivity and
 lowered electronic conductivity than pristine K by EISs, Tafel plots and D.c. conductivity
 measurements (see as Figure 3c, Figure 3d and Figure 3e). Meanwhile, we also characterized KF
 in the CCPP samples by XPS analysis (see Figure 2j), KF is a good SEI component that can
 provide high ionic conductivity and strong hardness (Ref. *Angew.Chem.Int. Ed.* **57**, 10864-10867
 (2018), Ref. *Adv. Funct. Mater.* **31**, 2005933 (2021), Ref. *Energy Environ. Sci.* **13**, 503-510
 (2020)).

Figure 3. (c) Nyquist plots taken at various cycle numbers.

Figure 3. (d) Tafel plots and derived exchange current density obtained from linear sweep voltammetry tests in the range of -0.3 V to 0.3 V at 2 mV s^{-1} .

**Figure 3.** (e) D.c. conductivity Measurements of pristine K and CCPP using blocking electrodes.

Figure 2. (j) C 1s, K 2p and F 1s XPS spectra of CCPP.

(2) In addition, in order to explore if the KF-based robust layer of CCPP can effectively avoiding electrolyte leakage into the Li surface, we conducted XRD characterization of CCPP after full contact with 3 M KFSI/EC:DEC electrolyte and EC:DEC solvent (Figure R10/R11). The results show that no reaction components of underlying K with electrolyte/ solvent in the CCPP samples (Figure R10a and Figure R11a), while KC_{24} , K_2CO_3/K_2CO_3 were evident in the pristine K (Figure R10b and Figure R11b), indicating that KF-based robust layer can prevent electrolyte penetration.

Figure R10. The XRD patterns of CCPP (a) and pristine K (b) after full contact with EC:DEC

solvent.

Figure R11. The XRD patterns of CCPP (a) and pristine K (b) after full contact with 3 M

KFSI/EC:DEC electrolyte.

**1-b:** Reviewer comments “The authors should add the results of symmetric and K/PBA
cells filled with the most common KPF6/EC:DEC electrolyte to show that the electrochemical
properties are independent of the electrolyte.”

**Response to 1-b:** In order to prove the electrochemical properties are independent of the
electrolyte. We added the results of symmetric and full cells filled with the most common
KPF6/EC:DEC electrolyte according to your suggestion, the results show that CCPP possess
better performances not only in the highly concentrated KFSI/EC:DEC electrolyte but also in
KPF6/EC:DEC electrolyte.

In the MS, page 14, line 308-321, we added: In addition, the CCPP anodes also exhibits
remarkable superiority in KPF6/EC:DEC electrolyte. Under a large current density of 2 mA cm^{-2} ,
maintaining a stable polarization for more than 110 h (Figure 4c). In contrast, DNPP and pristine
K become short circuit after 56 h and 35 h respectively, due to the lack of stable and robust SEI
protection. Figure 4d shows the voltage profiles at 0.5 mA cm^{-2} and 0.5 mAh cm^{-2} , pristine K
and DNPP short circuit ascribe to the dendrites within 90 h and 140 h, respectively. CCPP, by
contrast, can run stably more than 250 h, which is almost 3 times that of pristine K. For the rate
performance, CCPP can withstand circulation at higher current densities (0.1 mA cm^{-2} to $0.5, 1,$

1.5, 2 and 3 mA cm⁻²). Whereas, pristine K cannot last more than 4 cycles at 1.5 mA cm⁻² and 1
mAh cm⁻² (Figure S23). In a word, CCPP possess better performances not only in the highly
concentrated KFSI/EC:DEC electrolyte but also in KPF₆/EC:DEC electrolyte, which indicate
that the importance of a continuous and compact robust SEI to inhibit K metal dendrites.

In the MS, page 16, line 365-383, we added: Tests of full cells matched with PTCDA
confirmed the feasibility of CCPP anodes in real applications. Figure 6a shows the long-term
stability of CCPP||PTCDA and pristine K||PTCDA at the current density of 500 mA g⁻¹, the
CCPP||PTCDA cell shows a discharge capacity of 133.56 mAh g⁻¹ (5st cycle) when stabilized for
a number of cycles and keep a capacity of 117.5 mAh g⁻¹ after 200 cycles with a capacity
retention of 88% (Figure 6b). However, pristine K||PTCDA is broken soon after 31 cycles
because of the intensified dendrites growth and electrolyte decomposition for the lack of stable
SEI protection. Furthermore, the CCPP||PTCDA cell delivers reversible capacities of 138.8,
136.8, 134.6, 132.6, 130, 120 mAh g⁻¹ at current density of 30, 50, 100, 200, 500, 1000 mA g⁻¹,
respectively (Figure 6c and Figure S28). By contrast, pristine K||PTCDA cell deliver 138.4, 135,
131.5, 127.2, 115, 103.6 mAh g⁻¹, respectively. These results direct to superb rate capability of
CCPP||PTCDA cell. In addition, CCPP||PB and pristine K||PB cells were also assembled and
tested to verify the effects of the continuous and compact SEI. As shown in Figure S29, although
both cells deliver comparable initial capacity of about 77 mAh g⁻¹, the capacity of pristine K||PB
drops to 67.7 mAh g⁻¹ after 150 cycles (Figure S29c). The CCPP||PB cell, in contrast, could
deliver a higher capacity of 74.9 mAh g⁻¹ (Figure S29b), indicating reduced loss of electrolyte
and capacity of CCPP||PB cell.

In the MS and SI, we have added the following figures:

Figure 4. (c) Polarizations of CCPP, DNPP and pristine K at 2 mA cm⁻² and 0.5 mAh cm⁻² by
using KPF₆/EC:DEC electrolyte.

Figure 4. (d) Polarizations of CCPP, DNPP and pristine K at 0.5 mA cm^{-2} and 0.5 mAh cm^{-2} by
 using $\text{KPF}_6/\text{EC}:\text{DEC}$ electrolyte.

Figure S23. Rate performances of CCPP and pristine K at different current densities in the
 $\text{KPF}_6/\text{EC}:\text{DEC}$ electrolyte.

Figure 6. (a) Long-term stability of CCPP||PTCDA and pristine K||PTCDA batteries at 500 mA
 g^{-1} .

**Figure 6. (c)** The rate performances of CCPP||PTCDA and pristine K||PTCDA cells.

**Figure S29. (a)** Long-term cycling performance of CCPP||PB and pristine K||PB cells at 100 mA
g^{-1} by using KPF₆/EC:DEC electrolyte.

**2、Review comments:** Figure 3c and Figure S16. The scales of the x and y axes of the Nyquist
plot should be set equal.

**Response:** Thank you very much for your valuable suggestion. According to your suggestion, we
set the scales of the x and y axes of the Nyquist plot in Figure 3c and Figure S16 to be equal.

In the MS (page 10) and SI (page 10), we redraw Figure 3c and Figure S16 by setting the
scales of the x and y axes of the Nyquist plot to be equal.

**Figure 3. (c)** Nyquist plots taken at various cycle numbers.

Figure S16. EIS measurements at different temperatures for (a) pristine K and (b) CCPP.

**3、 Review comments:** K/PB cell should be charged to a higher voltage (~4.2 V) because PB
 might show additional plateau attributed to low spin Fe²⁺/³⁺ at ca 4.0 V.

**Response:** Thank you very much for your valuable suggestion. Due to the inherent properties of
 the previous commercial PB, it brings low capacity and voltage platform in the full battery. As a
 result, we replaced the previous commercial PB with synthetic PB made by our lab. In the Figure
 S29, we charged K||PB cell to a higher voltage (~4.2 V).

In the SI, page 19, we added:

**Figure S29.** (a) Long-term cycling performance of CCPP||PB and pristine K||PB cells at 100 mA
 g⁻¹ by using KPF₆/EC:DEC electrolyte. (b) The 1st and 150th charge-discharge profiles of
 CCPP||PB cells at 100 mA g⁻¹. (c) The 1st and 150th charge-discharge profiles of pristine K||PB
 cells at 100 mA g⁻¹.

**4、 Review comments:** (Experimental) The following can be a typo. The PTFE film (20 mm)

**Response:** We apologize for this mistake. In the MS, page 18, line 411, we corrected “The PTFE
film (20 mm)” to “**The PTFE film (20 μm)**”.

**5、 Review comments:** The approximate weight of the K metal electrodes should be mentioned
because the capacity balance of K metal and PB is essential for the cycling performance of K/PB
cells.

**Response:** Thank you very much for highlighting this important issue. The mass of the
potassium foil is about 0.055 g.

In the MS, page 19, line 419, we added “**the mass of the potassium foil is about 0.055 g**”.

REVIEWERS' COMMENTS

Reviewer #1 (Remarks to the Author):

The authors have thoroughly revised the manuscript with detailed discussion to address the reviewer's concerns. I am satisfied with most revisions, but there remain some suggestions further promoting the work quality.

1. The XRD patterns of PTCDAs and Prussian blue should be provided.
2. In terms of the preparation of Prussian blue (PB) electrode, why FeCl₃ was used rather than FeCl₂? PB cathodes have great influence on full cell performance. The quality of PB materials is highly correlated to the synthetic routes. For example, chelate-assisted co-precipitation method is always recommended for the synthesis of K-PB with Fe(II) ions (ACS Energy Lett. 2017, 2, 1122; Nature Commun. 2021, 12, 2167), while Fe(III) is not a typical precursor for the synthesis of PB cathodes.
3. Further discussion is suggested about the phase information of CCPP anodes upon K plating/stripping.

Reviewer #3 (Remarks to the Author):

The manuscript is revised accordingly with consideration of all comments. The manuscript should be acceptable for publication.

The point-by-point response to the reviewers' comments.

Reviewer #1

The authors have thoroughly revised the manuscript with detailed discussion to address the reviewer's concerns. I am satisfied with most revisions, but there remain some suggestions further promoting the work quality.

Response: Thank you very much for the positive assessments on our work. We have done more work and made further revisions in the manuscript. Our responses to the points are as follows:

Comments 1. The XRD patterns of PTCDA and Prussian blue should be provided.

Response: Thanks for your suggestions. We have provided the XRD patterns of PTCDA and Prussian blue and added in the SI part as **Figure S30** and **Figure S31**. The results show that the commercial PTCDA and self-made Prussian blue are consistent with the crystal structure of PTCDA (Reference. Energy Storage Mater. 2020, 31, 318-327) and PB ($\text{Fe}_4(\text{Fe}(\text{CN})_6)_3$ PDF#01-0239).

Figure S30. XRD patterns of PTCDA powder.

Figure S31. XRD patterns of Prussian blue powder.

Comments 2. In terms of the preparation of Prussian blue (PB) electrode, why FeCl_3 was used rather than FeCl_2 ? PB cathodes have great influence on full cell performance. The quality of PB materials is highly correlated to the synthetic routes. For example, chelate-assisted co-precipitation method is always recommended for the synthesis of K-PB with Fe (II) ions (ACS Energy Lett. 2017, 2, 1122; Nature Commun. 2021, 12, 2167), while Fe (III) is not a typical precursor for the synthesis of PB cathodes.

Response: We thank you for raising useful comments. In our last version based on the suggestion from reviewer 2 (ACS Nano 15, 9167-9175 (2021)), we prepared the Prussian blue (PB) electrode to improve the performance of the full cell. In this preparation method, FeCl_3 was used as the precursor. The results show that the use of this prepared PB does increase the capacity of full cell to more than 70 mAh g^{-1} . Most important of all, the results proved that CCPP||PB cell possessed of superiority to pristine K||PB cell (Figure S29a), indicating our strategy can effectively protective the K metal anode.

Based on reviewer 1's suggestion, in order to verify whether Fe (II) ions as the precursor of PBA (Prussian blue analogue) cathode can further improve the performance of the full cell, we synthesized K-FeHCFe cathode by using FeCl_2 (Reference. J. Mater. Chem. A 2017, 5, 4325-4330). As presented in Figure R1, the K||K-FeHCFe full cells deliver higher capacity (around 120 mAh g^{-1} of initial capacity), however, the stability is not satisfied (decayed by nearly 30 mAh g^{-1} after only 50 cycles) at the same mass loading ($\sim 1.5 \text{ mg cm}^{-2}$), same electrolyte

(KPF₆/EC:DEC electrolyte) and same current density conditions (100 mA g⁻¹).

Fortunately, no matter what kind of cathode is matched, the full cell with CCPP has better electrochemical performance than the pristine K. All of this proved that our strategy is an effective way for the protection of high-energy density K metal anodes.

We also tried to synthesize cathode according to the recommendations (Ref. ACS Energy Lett. 2017, 2, 1122 and Ref. Nat. Commun. 2021, 12, 2167). Due to the time limited, we can only show the results based on our previous cathode by using FeCl₂ (Figure R1). We will apply these recommended cathodes in our next work. In the MS, we have cited the following references as the reviewer mentioned: line 30, we have cited Ref. ACS Energy Lett. 2017, 2, 1122 as Ref 5; line 30, we have cited Ref. Nature Commun. 2021, 12, 2167 as Ref 4.

Figure S29. (a) Long-term cycling performance of CCPP||PB and pristine K||PB cells at 100 mA g⁻¹ by using KPF₆/EC:DEC electrolyte.

Figure R1. Long-term cycling performance of CCPP||K-FeHCFE and pristine K||K-FeHCFE cells at 100 mA g⁻¹ by using KPF₆/EC:DEC electrolyte.

Comments 3. Further discussion is suggested about the phase information of CCPP anodes upon K plating/stripping.

Response: Thanks for your valuable suggestions. In order to study the phase information of CCPP anodes upon K plating/stripping, semi-in-situ XRD was performed as **Figure R2**. CCPP anodes mainly contain K, KF and C species during the cycling. Meanwhile, the intensity of K signal increases in the K plating process and C signal increases in the stripping process.

Figure R2. Semi-in-situ XRD patterns of CCPP anodes upon K plating/stripping under 0.5 mA cm^{-2} and 0.5 mAh cm^{-2} in the in the highly concentrated KFSI/EC:DEC electrolyte.

Reviewer #3

Comment. The manuscript is revised accordingly with consideration of all comments. The manuscript should be acceptable for publication.

Response: We appreciate the reviewer for approving the publication of our work. We also wish to thank the reviewer's constructive comments for improving the quality of our work.